ecology, genetics, environmental science

deep-sea mining, nematoda, metabarcoding, community structure, phylogeny

**Author for correspondence:**
Lara Macheriotou
e-mail: lara.macheriotou@ugent.be

# Phylogenetic clustering and rarity imply risk of local species extinction in prospective deep-sea mining areas of the Clarion–Clipperton Fracture Zone

Lara Macheriotou[1], Annelien Rigaux[1], Sofie Derycke[1,2] and Ann Vanreusel[1]

[1]Marine Biology Research Group, Department of Biology, Ghent University, Krijgslaan 281, Building S8, 9000 Ghent, Belgium
[2]Aquatic Environment and Quality, Institute for Agricultural and Fisheries Research (ILVO), Ankerstraat 1, 8400 Oostende, Belgium

LM, 0000-0002-5662-5689

An understanding of the forces controlling community structure in the deep sea is essential at a time when its pristineness is threatened by polymetallic nodule mining. Because abiotically defined communities are more sensitive to environmental change, we applied occurrence- and phylogeny-based metrics to determine the importance of biotic versus abiotic structuring processes in nematodes, the most abundant invertebrate taxon of the Clarion–Clipperton Fracture Zone (CCFZ), an area targeted for mining. We investigated the prevalence of rarity and the explanatory power of environmental parameters with respect to phylogenetic diversity (PD). We found evidence for aggregation and phylogenetic clustering in nematode amplicon sequence variants (ASVs) and the dominant genus *Acantholaimus*, indicating the influence of environmental filtering, sympatric speciation, affinity for overlapping habitats and facilitation for community structure. PD was associated with abiotic variables such as total organic carbon, chloroplastic pigments equivalents and/or mud content, explaining up to 57% of the observed variability and providing further support of the prominence of environmental structuring forces. Rarity was high throughout, ranging from 64 to 75% unique ASVs. Communities defined by environmental filtering with a prevalence of rarity in the CCFZ suggest taxa of these nodule-bearing abyssal plains will be especially vulnerable to the risk of extinction brought about by the efforts to extract them.

## 1. Introduction

The abyssal plains of the Clarion–Clipperton Fracture Zone (CCFZ) in the northeastern Equatorial Pacific (figure 1) contain the densest known aggregation of polymetallic nodules; mineral concretions abundant in commercially important metals (e.g. Ni, Cu, Co). Due to the increasing global demand for these metals in high-tech industries (e.g. electric car batteries, smartphones etc.), geopolitical matters potentially limiting their availability and the depletion of large, high-grade ore deposits, deep-sea mining of polymetallic nodules in the CCFZ is emerging as an alternative to land-based extraction [1]. Spanning 4.5 million km² between Mexico and the Hawaiian islands, this vast deep-sea ecosystem (approx. 4–5 km depth) is oligotrophic with an eastward increasing particulate organic carbon (POC) gradient [2] and remains largely unexplored, with most biological studies limited to a single licence area [3–6]. Concurrently, the mechanisms underlying community assembly in the CCFZ are all the more elusive as investigations of phylogenetic structure have rarely been applied to the deep sea [7–9]. The inclusion of phylogenetic

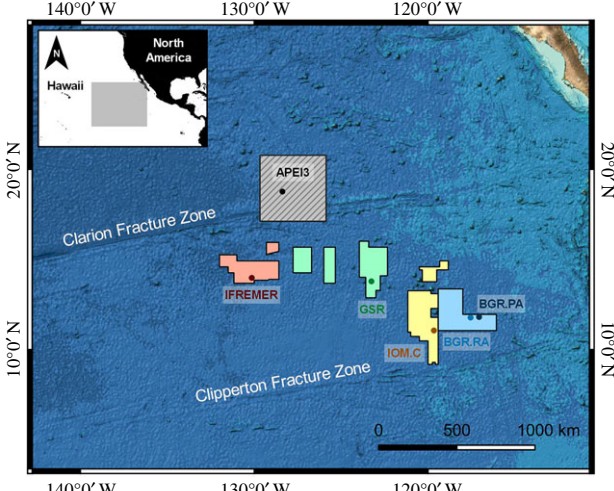

**Figure 1.** Areas sampled in the Clarion–Clipperton Fracture Zone (BGR, Bundesanstalt für Geowissenschaften und Rohstoffe; RA/PA, Reference/Prospective Area, respectively; IOM.C, InterOcean Metal-Control; GSR, Global Seabed Resources, IFREMER; Institut Français de Recherche pour l'Exploitation de la Mer; APEI3, Area of Particular Environmental Interest 3). Circles indicate sampling location within each area.

data enables the exploration of the nature of species interactions, their causality, and how these affect the ecological and evolutionary dynamics of taxa.

The establishment of species in a community is thought to be mediated by (i) neutral forces in which taxa are ecologically equivalent and thus persistence is governed by stochastic processes (e.g. dispersal), (ii) historical factors (e.g. starting conditions) which predominate over local dynamics and (iii) niche-related processes [10,11]. Species pairs exhibit positive associations when they co-occur more often than expected by chance, forming 'aggregations'. Conversely, 'segregations' develop when species pairs are found to co-occur less than expected by chance. These deviations from randomness are thought to be the result of species interactions, habitat preferences and/or speciation processes [12,13]. Moreover, competition is expected to be strongest between closely related and thus ecologically similar taxa, limiting their co-occurrence, termed the 'competition-relatedness hypothesis' [14,15]. From a phylogenetic perspective, co-occurring species being more closely related than would be expected by chance (clustering) results from environmental filtering due to one or more shared conserved trait(s), which allow them to persist in that locality in the absence of resource limitation and associated competitive interactions (although see [16]). The 'environment' in this instance consists of the abiotic characteristics of the location (e.g. elevation, grain size, temperature, geochemistry) as well as microscopic biotic components such as bacteria, viruses and fungi, while the relevant traits are those conferring persistence in that particular locality (e.g. body shape, the presence of precloacal supplements, buccal cavity). Alternatively, locally co-occurring species being less closely related to each other than expected by chance (overdispersion) results from competitive exclusion of conserved traits and/or convergent evolution of the traits defined by the environmental filter. Random structuring is thought to arise via local exclusion of convergent traits [17]. Consequently, assemblages that exhibit environmental filtering are thus more likely to show high sensitivity to changes in the environment than those structured mainly by intra- and interspecific interactions.

Rare species, which are inherently at a higher risk of extinction [18], are common in morphological assessments of deep-sea infaunal taxa, elevating biodiversity estimates and contributing substantially to variability in community composition [5,19,20]. Given the predicted extent of deep-sea mining impacts on the CCFZ biota [21], the prevalence of rare taxa, and the (mostly undescribed) high species richness of abyssal invertebrates in general, it becomes crucial to understand community assembly processes and how these contribute to the maintenance of diversity in the context of mining mitigation. Here, we focus on free-living marine nematodes which are found in deep-sea sediments worldwide [22] and constitute the most abundant benthic invertebrate phylum in the CCFZ [3,5,23,24]. We apply a metabarcoding approach to gain insights regarding the following questions:

— Is the distribution of deep-sea nematodes in the CCFZ non-random with respect to co-occurrence (aggregation/ segregation) and phylogeny (clustering/overdispersion)? If so, what are the tentative mechanisms defining community structure?
— How does phylogenetic diversity (PD) vary across the CCFZ? Are the observed patterns attributable to environmental characteristics (e.g. POC gradient)?
— Is rarity a dominant feature of CCFZ nematode assemblages with respect to genetic variability (ASVs)?

## 2. Material and methods

### (a) Sample collection

Sediment cores were collected during the EcoResponse campaign (Assessing the Ecology, Connectivity and Resilience of Polymetallic Nodule Field Systems) aboard the German Research Vessel SONNE from 11 March 2015 to 30 March 2015. Four license areas across 1300 km were targeted (figure 1; BGR, Bundesanstalt für Geowissenschaften und Rohstoffe; IOM.C, InterOcean Metal-Control; GSR, Global Seabed Resources; IFREMER, Institut Français de Recherche pour l'Exploitation de la Mer) and one Area of Particular Environmental Interest (APEI3). The BGR area was sampled at two sub-regions; the 'Reference area' and 'Prospective area' (BGR.RA: limited mining, BGR.PA: intensive future mining). Samples were collected using the Octopus multicorer (inner diameter: 94 mm) at 23 stations (electronic supplementary material, table S1). Replicates of each area were retrieved from separate multicorer deployments (1 core/deployment, BGR.PA: $n = 5$, BGR.RA: $n = 4$, IOM.C: $n = 2$, GSR: $n = 4$, IFREMER: $n = 5$, APEI3: $n = 3$). The top 5 cm of each core were sliced and frozen in sealed plastic bags at −20°C. Meiofauna was extracted from half of the sediment into sterile Milli-Q water by density-gradient centrifugation (3 × 12 min, 1905 rcf) with the colloidal silica polymer Ludox HS-40 as a flotation medium (specific density 1.18 g cm$^{-1}$) [25]. Cores for environmental variables were sliced (0–1 cm and 1–5 cm) and stored at −20°C; a 1 ml was subsampled from the 0–1 cm layer and stored at −80°C for pigment analysis (chlorophyll-a, phaeopigments). Extraction was completed using 90% acetone and concentrations were measured fluorometrically using the Trilogy Laboratory Fluorometer at the Max Planck Institute for Marine Microbiology. Grain size analysis was completed using laser diffraction (Malvern Mastersizer Hydro 2000 G). Total organic carbon (TOC) and total nitrogen (TN) was quantified by combustion of freeze-dried samples with the Flash 2000 NC Sediment Analyser.

## (b) Sequence data analysis

A full description of DNA extraction and HTS library preparation protocols are available in electronic supplementary material. Gene-specific adapter sequences were truncated from the 5′ and 3′ read ends using Cutadapt [26] (v. 1.12; electronic supplementary material, 'Cutadapt output'). ASVs were generated with DADA2 [27] under default settings with the exception of truncation of forward and reverse reads at 225 and 250 bp, respectively (electronic supplementary material, 'DADA2 output'). Taxonomic assignment of ASVs was completed in QIIME1 [28] (assign_taxonomy.py) with the Naive Bayesian RDP [29] classifier (confidence estimate: 0.80) in two steps: first, a large eukaryotic reference training set was used (Silva release 123 for QIIME1, 99% OTUs and UGent nematode Sanger sequences; $n = 20\,201$). Second, all ASVs that received a 'Nematoda' label were extracted; to these taxonomic assignment was completed using a smaller, nematode-exclusive training set (2178 sequences, 27 Orders). Samples were rarefied to the lowest sequence count ($n = 25\,258$) for subsequent analyses (QIIME1, alpha_rarefaction.py). ASVs lacking a taxonomic assignment beyond Nematoda were included in our analyses to investigate whether these (presumably) unknown taxa exhibit patterns similar to those identified at genus-level.

## (c) Diversity analyses

Normality and homoscedasticity of the number of genera assigned to Nematoda ASVs were assessed with the Shapiro–Wilk and Levene test respectively while statistically significant differences were determined via analysis of variance (ANOVA). Normality could not be determined for IOM.C due to insufficient replication. Shared/unique Nematoda ASV plots were generated with the R package 'UpSetR' [30]. Phylogenetic beta-diversity was assessed with the unweighted UniFrac [31] distance in Nematoda ASVs and visualized by means of principal coordinates analysis (PCoA).

## (d) Community structure analyses

We tested for non-random co-occurrence patterns at the regional (CCFZ) and local (area) level in Nematoda ASVs (= Genus-assigned + Unassigned, $n = 1981$) and within the three most abundant genera: *Acantholaimus* ($n = 234$), *Desmoscolex* ($n = 286$) and *Halalaimus* ($n = 142$). Where possible analyses were completed with replicates pooled and unpooled to determine whether patterns are consistent at a finer spatial scale:

CCFZ.Replicates: data matrix includes all areas, replicates not pooled (i.e. 23 columns).
CCFZ.areas: data matrix includes all areas, replicates pooled by area (i.e. six columns).
Local: data matrix includes replicates of each area separately (i.e. six data matrices, analysis cannot be completed with replicates pooled).

The number of Checkerboard Pairs (CPs) and C-score [32] were compared against a null model (sim2, 10 000 replicates) in the R package 'EcoSimR' [33]. The sim2 algorithm randomizes a presence-absence matrix by reshuffling elements within each row equiprobably. The C-score quantifies the average number of CPs across all possible paired combinations; in a matrix with sites as columns and taxa as rows, for each unique pair, the C-score is equal to $C_{ij} = (R_i - S)(R_j - S)$ where $R_i$ and $R_j$ are row sums of taxa $i$, $j$ and $S$ is the number of shared sites in which both $i$ and $j$ are present. Thus, for any particular species pair, the C-score is a numerical index that ranges from a minimum of zero (maximally aggregated) to a maximum of $R_i \times R_j$ (maximally segregated with no shared sites). Departures from randomly co-occurring assemblages were assessed via the

standardized effect size (ses, see below) as well as upper- and lower-tail one-sample *t*-tests. An assemblage that is structured mainly by competitive interactions (segregation) is expected to have more CPs and higher C-scores than would be expected by chance (i.e. ses > 0) [13].

Reads were aligned with the R package 'DECIPHER' [34] using a chained guide tree; alignments were subsequently adjusted with the adjust.alignment command. The software 'ModelTest-ng' [35] was used to determine the appropriate model of nucleotide evolution for tree construction for each ASV set. In all instances the general time reversible (GTR) model was selected according to the Bayesian and/or the Akaike information criterion. Approximately maximum-likelihood phylogenetic trees were constructed using FastTree [36] under the GTR + CAT model; midpoint rooting was completed with the R package 'phangorn' [37]. PD (see below) and ses of the following phylogenetic metrics were quantified for Nematoda, Genus-assigned ($n = 978$), Unassigned ($n = 1003$), *Acantholaimus*, *Desmoscolex* and *Halalaimus* ASVs using the R package 'picante' [38].

—PD [39]: sum of branch lengths between the root and all species in a sample phylogeny.
—Mean Pairwise Distance (MPD [40]): mean pairwise phylogenetic distance between all taxa in a local assemblage.
—Mean Nearest Taxon Distance (MNTD [40]): mean phylogenetic distance between each taxon and its nearest neighbour on the phylogenetic tree with which it co-occurs in the local assemblage.

The observed metric (obsMetric) was compared to that obtained from 999 randomizations (nullMetric) of the assemblage generated using all null models (i.e. taxa.labels, richness, frequency, sample.pool, phylogeny.pool, independentswap, trialswap, see electronic supplementary material, table S12 for full description). The ses was calculated as follows:

$$\text{ses.Metric} = \frac{(\text{obsMetric-nullMetric})}{\text{stdev.nullMetric}},$$

where stdev equals the standard deviation; negative ses values indicate phylogenetic clustering while positive values suggest phylogenetic overdispersion [41]. Statistically significant departures from a zero mean for ses.PD/ses.MPD/ses.MNTD at $\alpha = 0.05$ were tested using a one-sample two-tailed *t*-test when normally distributed as determined by the Shapiro–Wilk test. Alternatively and in the event of fewer than three replicates per area (i.e. IOM.C and *Desmoscolex* ASVs for APEI3), the Wilcoxon signed-rank test was used. These metrics have commonly been used to reveal phylogenetic structure in bacterial, protist, plant, bird and mammal assemblages [40–45].

Linear regressions were performed to investigate the relationship between (independent) environmental variables (%total organic carbon [%TOC], %Mud [less than 4 μm] and chloroplastic pigment equivalent [CPE, $\Sigma$ chlorophyll-a + phaeopigments]) and (dependent variable) PD for Nematoda, Genus-assigned, Unassigned, *Acantholaimus*, *Desmoscolex* and *Halalaimus* ASVs using the 'lm' function. Homoscedasticity of the independent variables was assessed with the Levene test; %Mud was square-root transformed. Multicollinearity between independent variables was assessed with the Pearson correlation coefficient and the variance inflation factor (VIF < 10 indicating absence of multicollinearity); these indicated that the correlation was fair to moderate [46] and within the permissible range for the subsequent regression analyses. In all instances the full model including 2-way and three-way interactions (PD ∼ TOC + CPE + Mud.sqrt + TOC: CPE + TOC:Mud.sqrt + CPE:Mud.sqrt  TOC:CPE:Mud.sqrt) was used as starting point. Terms with the highest *p*-values were sequentially removed to arrive at the final model including only

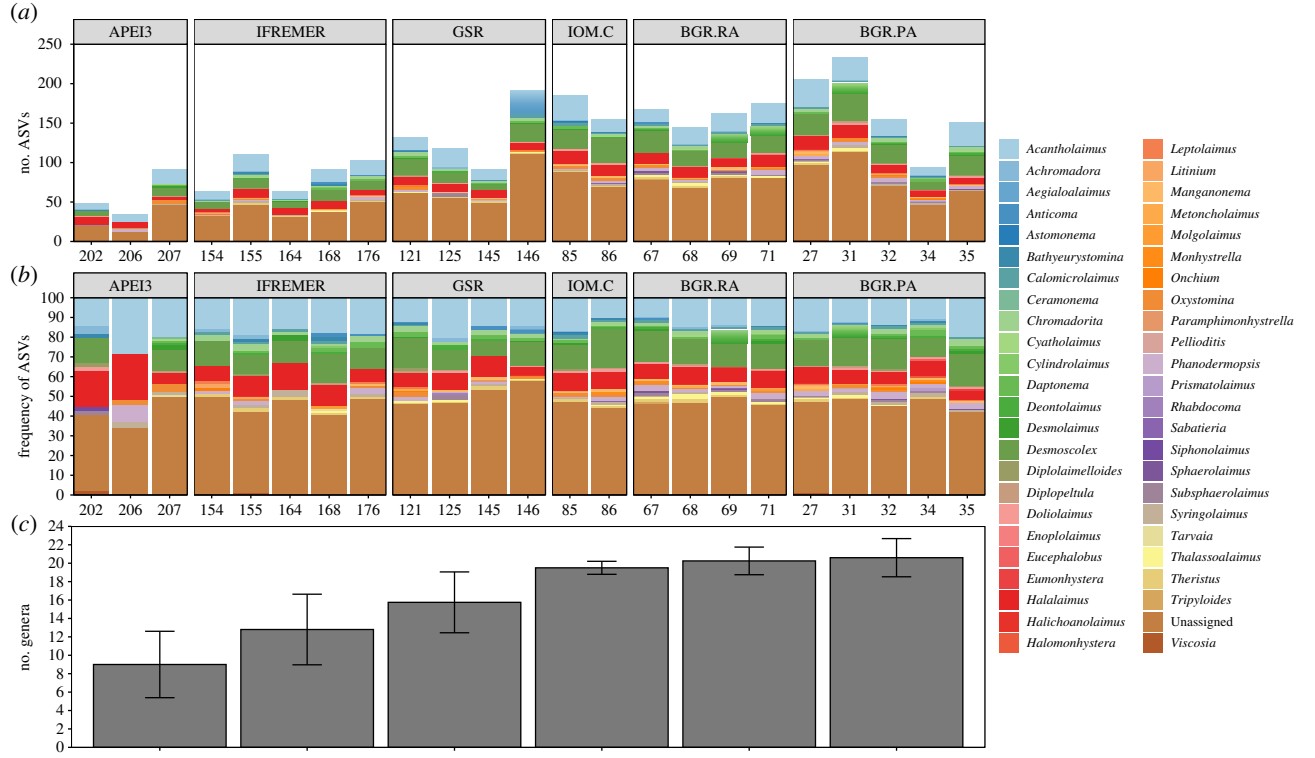

**Figure 2.** Taxonomic assignments and generic richness of Nematoda ASVs. (*a,b*) Absolute/relative abundance of genus-level taxonomic assignments of rarefied Nematoda ASVs for each replicate sampled in the BGR.PA, BGR.RA, IOM.C, GSR, IFREMER and APEI3 areas. (*c*) Generic richness of ASVs in each area, error bars represent standard deviation.

statistically significant terms. Outliers (defined as data points whose absolute value exceeded three times the standard deviation) were removed and normal distribution of the residuals assessed with the Shapiro test *a posteriori*. Unless indicated otherwise, all analyses were completed in RStudio. A diagram of our sampling design and analysis is provided in electronic supplementary material, figure S1.

## 3. Results

The 23 rarefied samples generated 5574 ASVs of which a large fraction (41%) were 'Unassigned', 1981 were assigned to 'Nematoda' (35%), followed by 'Arthropoda' (12%) with each of the remaining 33 phyla being represented by less than 2% relative abundance (electronic supplementary material, figure S2). The most abundant nematode genera were *Desmoscolex*, *Acantholaimus* and *Halalaimus* with 286, 234 and 142 ASVs, respectively, while the remaining 44 were represented by less than 40 ASVs each (figure 2). These three genera were also the most widespread with *Acantholaimus* dominant (relative abundance greater than 5%) in all 23 samples, *Desmoscolex* in 22 and *Halalaimus* in 21. One ASV assigned to *Acantholaimus*_sp., in particular, was virtually ubiquitous throughout the CCFZ, occurring in all but one replicate. Only 12 nematode genera in total were restricted to a single area (2–4 per area), while 13 (28%) were common to all six.

### (a) Nematode assemblages

Generic richness of genus-assigned ASVs differed significantly between areas ($p = 0.0001$) and was lowest in APEI3 (mean: 9.00, s.d.: ±3.61) and highest in BGR.PA (mean: 20.60, s.d.: ±2.07; figure 2; electronic supplementary material, tables S2 and S3). Pairwise comparisons indicated APEI3 had significantly fewer nematode genera than

IOM.C, BGR.RA, BGR.PA while IFREMER had significantly fewer genera than BGR.RA and BGR.PA (electronic supplementary material, table S4). Over 63% of Nematoda ASVs were restricted to one area, with APEI3 exhibiting over 75% unique ASVs (electronic supplementary material, figure S3). Just two Nematoda ASVs were ubiquitous in the CCFZ (*Acantholaimus*_sp., *Daptonema*_sp.) while the two areas nearest to each other (BGR.RA, BRA.PA) shared the largest number of ASVs ($n = 43$). Notably, five ASVs were shared between the most distant areas APEI3 and BGR.PA.

PD was consistently lowest in APEI3 and increased gradually to maximum values in IOM.C (figure 3; electronic supplementary material, table S5), exhibiting statistically significant differences between areas ($p = 0.0039$, electronic supplementary material, table S6). Pairwise comparisons indicated APEI3 had lower PD than IOM.C, BGR.RA and BGR.PA (electronic supplementary material, table S7). Unassigned ASVs had the greatest contribution to total PD. At the generic level, *Desmoscolex* exhibited the highest PD (mean: 0.733, s.d.: ±0.230) followed by *Halalaimus* (mean: 0.377, s.d.: ±0.095) and *Acantholaimus* (mean: 0.281, s.d.: ±0.072).

A correlation matrix and correlation chart of the environmental variables are included in electronic supplementary material, figure S4. The regression analyses indicated a differential importance of environmental variables in explaining PD within each ASV set (electronic supplementary material, table S8). In Nematoda ASVs, the combined effect of CPE, %TOC and their interaction accounted for 48% of the observed variability; the former two were positively related to PD while the latter negatively. TOC and CPE described 57% of the variability of Genus-assigned PD, with a negative and positive association respectively, while TOC content was positively related to PD in Unassigned ASVs, describing 33% of the variability. Mud content alone explained *ca* one-third

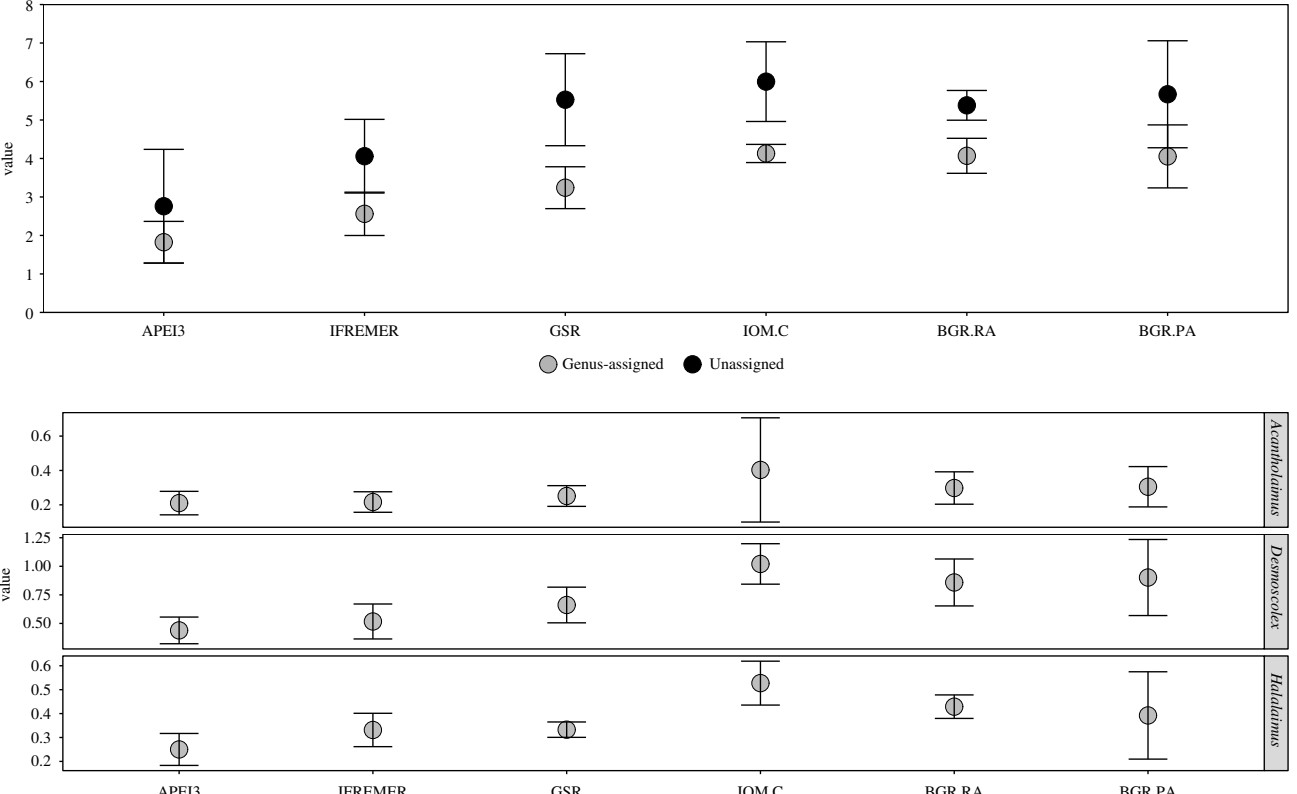

**Figure 3.** Phylogenetic diversity (PD) of Nematoda ASVs (composed of Genus-assigned (light grey) and Unassigned (dark grey), top panel), *Acantholaimus*, *Desmoscolex* and *Halalaimus* ASVs (bottom panel) in the BGR.PA, BGR.RA, IOM.C, GSR, IFREMER and APEI3 areas. Circles and error bars represent mean and standard deviation, respectively.

of the variability in *Halalaimus* ASVs and was negatively associated with PD; no significant models could be generated for *Acantholaimus* and *Desmoscolex* ASVs.

The topology of the UniFrac-based PCoA approximately matched the geographical distribution of the different areas in the CCFZ (electronic supplementary material, figure S5). The more eastern areas (BGR.PA, BGR.RA, IOM.C, GSR) exhibited a high degree of overlap with the two BGR areas being virtually superimposed. Contrastingly, the western areas IFREMER and APEI3 were most segregated. No statistical testing could be completed on this metric (PERMDISP < 0.05).

## (b) Patterns of co-occurrence

The number of CPs and C-score for Nematoda ASVs were lower than expected by chance at both the Regional and Local level for all areas excluding IOM.C and BGR.RA, the latter two exhibiting random co-occurrences and an excess of CPs (segregation), respectively (figure 4; electronic supplementary material, table S9). Moreover, patterns were consistent whether replicates were pooled or not. Co-occurrence patterns within the genera *Acantholaimus* and *Desmoscolex* were nearly identical, exhibiting fewer CPs and lower C-scores than expected by chance, which was observed at the Regional (CCFZ.Replicates, CCFZ.areas) and Local level in 3 and 4 out of six areas, respectively. Co-occurrence in *Halalaimus* ASVs was idiosyncratic; at the Regional level (CCFZ.Replicates) CPs and C-score did not differ from the null model, yet both metrics indicated higher than expected co-occurrences when replicates were pooled by area. Within each area, the number of CPs and C-score in GSR and BGR.RA were higher than expected by chance, while the remaining areas were random.

## (c) Phylogenetic community structure

The vast majority of ses.PD, ses.MNTD and ses.MPD were negative pointing to an increased tendency towards phylogenetic clustering (figure 5). Nematoda ASVs exhibited the most clustering across areas, as well as the highest consistency between the three metrics (electronic supplementary material, table S10). The strongest signal came from ses.MPD where all areas but IOM.C differed significantly from zero, followed by ses.MNTD in which all areas excluding GSR and IOM.C exhibited clustering, and finally, ses.PD in which IFREMER, and BGR.RA, BGR.PA were clustered (electronic supplementary material, table S11). In Genus-assigned ASVs, APEI3, IFREMER and IOM.C exhibited phylogenetic clustering in ses.PD and ses.MNTD while ses.MPD suggested the assemblages did not differ from the null model. Clustering in Unassigned ASVs was restricted to BGR.RA, BGR.PA (ses.PD/ses.MNTD) and BGR.RA with ses.MPD. Clustering was discordant between genera; *Desmoscolex* ASVs did not differ from a randomly structured assemblage in all areas and for all metrics while *Halalaimus* exhibited clustering only in APEI3 with ses.MPD while *Acantholaimus* ASVs were clustered in all areas excluding IOM.C, BGR.RA (ses.MPD), in APEI3, GSR, BGR.PA (ses.MNTD) and in IFREMER, GSR (ses.PD).

## 4. Discussion

### (a) Taxon co-occurrence, phylogenetic community structure and diversity

Overall nematode ASVs were characterized by aggregation, with CPs and C-scores that were lower than expected by

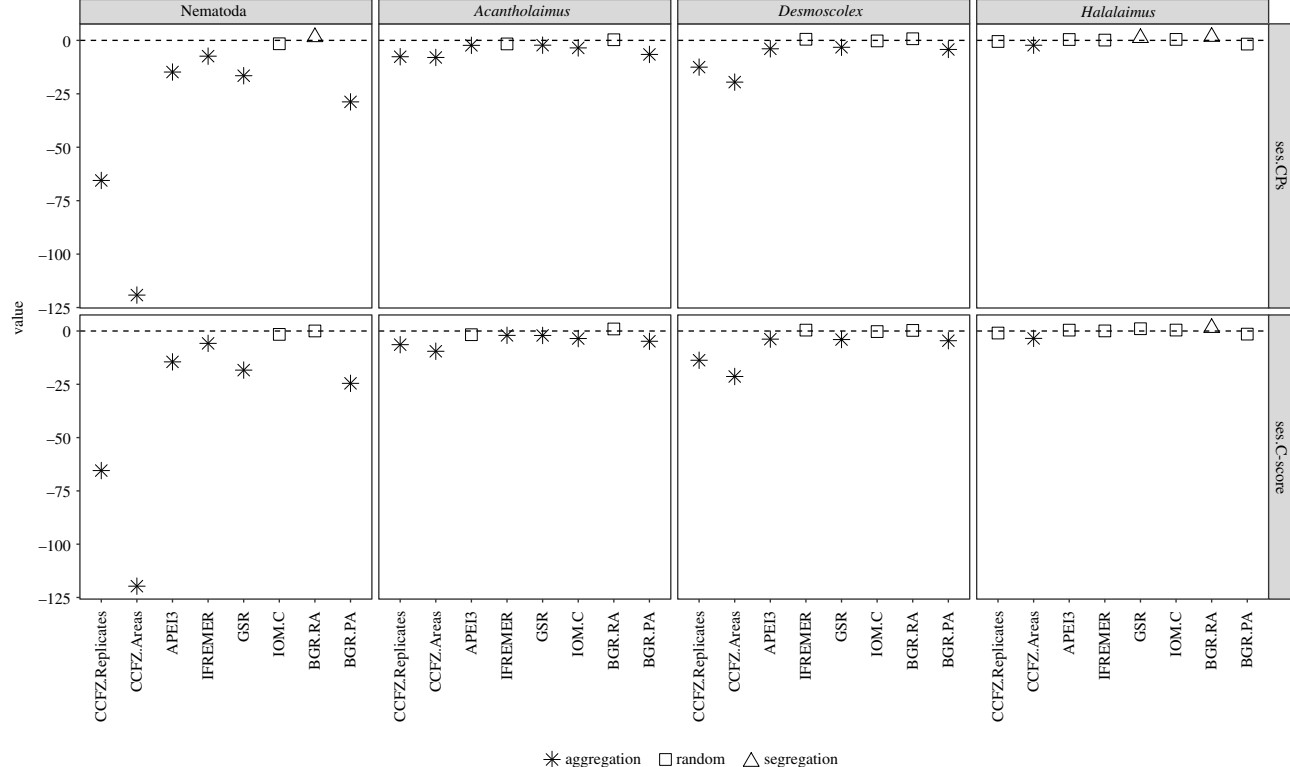

**Figure 4.** Principal coordinate analysis (PCoA) of unweighted UniFrac distance of Nematoda ASVs in the BGR.PA (light blue), BGR.RA (dark blue), IOM.C (yellow), GSR (green), IFREMER (red) and APEI3 (purple) areas.

chance, both at the regional level (CCFZ) and within each area in all ASV sets. Moreover, this was generally consistent at the scale of individual replicates, suggesting that small-scale habitat heterogeneity was minimal (electronic supplementary material, figure S6) with no effect on the overarching patterns of co-occurrence. Segregation, on the other hand, was very rarely observed in our data, despite being common in terrestrial nematodes [47], but also in microbial plankton, reptiles and birds [48]. Aggregated nematode assemblages in the CCFZ are most likely the result of facilitative interactions, affinity to similar habitats and/or sympatric speciation [13,48]. Given that anthropogenic disturbance can generate shifts in co-occurrence patterns [49], the investigation thereof offers an additional means for detecting the impacts of deep-sea mining which go beyond conventional community abundance and diversity.

From a phylogenetic perspective nematode assemblages in the CCFZ were characterized by relatedness that was higher than expected by chance (clustering), which has also been documented in deep-sea octocorals [7], peracarid crustaceans [8] and bacterioplankton [9]. Although analysis at finer taxonomic resolution has been shown to increase the signal of overdispersion [50,51], this was absent in our data. Clustering was predominantly observed in Nematoda, Genus-assigned and *Acantholaimus* ASVs, while *Desmoscolex* and *Halalaimus* were randomly structured. This differential response indicates that factors controlling phylogenetic structuring, or the relative dominance thereof, can differ between genera. Moreover, clustering was observed in ses.MPD, a 'basal' metric more sensitive to deeper (older) branching, as well as ses.PD/ses.MNTD which are 'terminal', best at detecting more recent processes at tree tips [52]. Thus, at each locality, increased phylogenetic similarity is the result of both historical and more recent evolution. Currently, two processes are considered to result in phylogenetic clustering: (1) environmental filtering of

conserved traits [17], and (2) competitive exclusion of conserved and/or convergent traits [16]. The proposition that deep-sea benthic communities are not structured by competition was originally explained with the 'stability-time hypothesis' [53] which attributed the coexistence of many species to the environmental stability of the deep-sea, allowing evolutionary processes to generate intense niche partitioning. Contemporary communities are thus the result of past competitive interactions. Competitive exclusion has been shown to be rare in shallow soft sediments, which if extended to the deep-sea realm can help explain its high diversity [54]. In addition, the oligotrophic nature of the CCFZ and the resulting low population densities may themselves inhibit competition [54]. To the extent to which such interactions would manifest in checkerboard pairs, we find little evidence to support competitive exclusion in defining nematode assemblages in the CCFZ. Moreover, the small size of each sample (347 cm³) and genus-level analysis, falls within the 'Darwin-Hutchinson zone', coined by Prof. Michael Donoghue (unpublished) which is the plane of taxonomic and spatial resolution where competition is expected to operate [55]. Concurrently, the combined effect of limited dispersal and genetic drift cannot be dismissed as an explanation for the observed clustering. Nonetheless, taken together, both our phylogenetic and co-occurrence data are in support of the long-held belief that deep-sea communities are predominantly defined by environmental characteristics [56–58] rather than biotic competitive interactions.

In the absence of competition, clustering can be attributed to environmental filtering by which closely related species coexist due to shared conserved trait(s) related to tolerance of the local conditions, dictated by said filter. Additional support for the deterministic influence of the environment on nematode assemblages was provided by the regression analysis showing that approximately half of the observed variability in PD could be

*Proc. R. Soc. B* **287**: 20192666

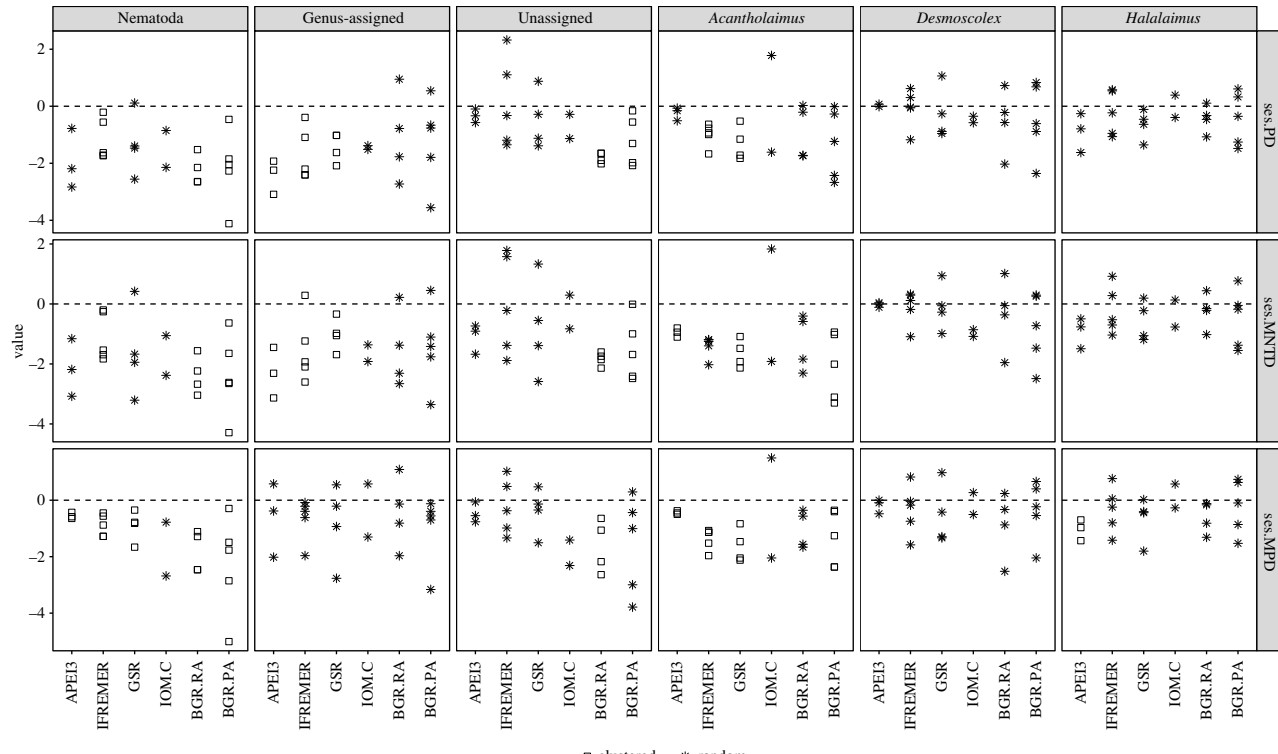

**Figure 5.** Standard effect size (ses) of phylogenetic diversity (ses.PD), mean nearest taxon distance (ses.MNTD) and mean pairwise distance (ses.MPD) of Nematoda, Genus-assigned, Unassigned, *Acantholaimus*, *Desmoscolex* and *Halalaimus* ASVs in the BGR.PA, BGR.RA, IOM.C, GSR, IFREMER and APEI3 areas. Symbols represent outcome of the relevant *t*-test: squares, asterisk indicate statistically significant clustering and random phylogenetic structure, respectively.

explained as a function of organic content, pigment concentrations and sedimentary characteristics for the different ASV sets. The increased availability of primary production (CPE) and its by-products (TOC) in the eastern CCFZ (electronic supplementary material, figure S6), was associated with higher PD values, suggesting that food availability leads not only to higher species richness but also to more phylogenetically variable assemblages. In this way, nematode communities in the eastern CCFZ could be considered food-limited, yet we maintain that this limitation does not promote intense competitive interactions between individuals.

Random co-occurrence patterns within *Desmoscolex* and *Halalaimus* ASVs were commonly associated with random phylogenetic structure, providing two lines of evidence to support the dominance of stochastic processes in these assemblages. Keeping in mind that at a given spatial scale communities are the result of several processes that are difficult to isolate [10], non-random phylogenetic structure could be obscured by opposing and nullifying forces, the inconsequentiality of phylogeny in defining these assemblages [45,59] and/or sampling limitations.

An additional striking difference between genera was average PD, which nearly doubled in *Desmoscolex* ASVs. Both *Acantholaimus* and *Halalaimus* are thought to have originated in the deep sea [60] whose relative stability and extreme conditions may be decelerating evolutionary rates and even limiting evolutionary opportunity [61,62]. Thus, it may be that *Desmoscolex* represents a successful, yet relatively recent shallow-water addition to the CCFZ fauna. PD values were comparable between Unassigned and Genus-assigned, suggesting the former were composed of multiple genera. Although similar PD could arise from any number of tree configurations resulting in nearly equal branch length sums, the most parsimonious explanation would be that ASV sets of approximately equal PD and ASV richness exhibit similar phylogenetic tree structure due to the presence of common genera.

## (b) Rarity in the CCFZ

All six areas showed a high degree of rarity represented by a large proportion of unique ASVs, collectively representing 85% of the entire ASV assemblage (electronic supplementary material, figure S3). Concurrently, one needs to interpret such patterns while taking into account sampling limitations and the spatio-temporal microhabitat heterogeneity of abyssal plains, one of the driving forces of deep-sea diversity [56]. Small-scale differences in environmental characteristics could result in a taxon being rare in one patch of sediment yet abundant in another adjacent to it; most ASVs were unique not only to each area but also to each replicate (electronic supplementary material, figure S7). Moreover, rarity is a combination of both abundance and occurrence, and can be classified into three types of geographical distribution: (1) low abundance species that are continuously distributed and widespread, (2) low abundance species occurring at scattered localities over a large area, and (3) species with such a restricted geographical range that are considered rare even though they may be abundant at each locality [63]. Intraspecific variation in copy number in combination with intragenomic variation of the 18S locus preclude a quantitative assessment of nematode assemblages [64–66]. Moreover, given the impediments to accurate taxonomic assignment of highly similar reads by RDP (following section) and the single-nucleotide resolution of the DADA2 algorithm [27], we expect some non-trivial fraction of unique and/or Unassigned ASVs to be attributable to this variation. Nonetheless, our data suggest that rarity of Nematoda ASVs, just as of nematode species [24], is a prominent feature of the CCFZ and characterized by both widely scattered and

highly localized taxa. The importance of rarity is inherently challenging to define; however, rare species have been shown to sustain high-diversity ecosystems by carrying the least redundant trait combinations, possibly representing the most vulnerable ecological functions [67]. Despite their small size, meiofaunal taxa, which in the CCFZ are represented primarily by nematodes, have a large effect on sedimentary properties which consequently influence various ecosystem services (e.g. denitrification, heavy metal removal) [68–70]. Our data suggest that one of the nine Areas of Particular Environmental Interest (APEI3), whose location was in part defined by its capacity to protect 'a full range of habitats' [71], differs most from the investigated license areas in terms of nematode assemblages and environmental characteristics. This fact undoubtedly renders it even more worthy of protection, while simultaneously stressing the need for a revaluation of the localities excluded from future mining on the premise of their presumed representativeness of areas that are destined to be mined. In light of the severe and, in some ways, irreversible impacts of these operations (e.g. sediment compaction and plume, nodule removal), as well as the increased vulnerability of rare species to extinction [18], we feel it is imperative to adopt the precautionary principle and to empirically assess the functionality and representativeness of the selected APEIs.

## (c) Metabarcoding nematode assemblages

Genus-assigned ASVs identified assemblages that were similar to morphological assessments [4–6,24,72] typically exhibiting a dominance of *Acantholaimus*, *Desmoscolex* and *Halalaimus*. Genera belonging to the family Monhysteridae, specifically *Thalassomonhystera* and *Monhystrella* which are known to be abundant in abyssal plains including the CCFZ [22,24], were a notable exception as these were absent and represented by a single ASV respectively. Comparatively generic richness was reduced which, in combination with the large proportion of Unassigned ASVs, highlights bottlenecks encountered in the analysis of HTS data. First is the inevitable bias introduced by the chosen primer set, which may fail to amplify certain species; the 22_R primer does in fact have a $C \rightarrow T$ mismatch at the 3rd base pair $(5' \rightarrow 3')$ in *Monhystrella* sequences. Although DNA transitions are generally more benign than transversions, single nucleotide mismatches can have severe impacts on the effectiveness of PCR amplification, especially when located in the 3' region [73], thus strongly reducing the relative amount of DNA available for sequencing. Second, the choice of taxonomic assignment method can influence which taxa are identified [74]. Both *Monhystrella* and *Thallassomonhystera* belong to the polyphyletic order Monhysterida [75], which includes the Xyalidae family, represented by 40 ASVs in our assemblage (*Theristus*: 14, *Manganonema*: 14, *Daptonema*: 12). In addition, 250 of the 1003 Unassigned ASVs lacked a taxonomic label beyond 'Monhysterida'. To test the effect of RDP we re-assigned taxonomy with a reference database excluding all Monhysterida sequences except *Monhystrella*, which increased the abundance of the latter from a single ASV to 82, making it the fourth most abundant genus. Taken together, these facts suggest that the reduced abundance/absence of *Monhystrella* and *Thalassomonhystera* in our data is primarily due to the degree of sequence similarity within Monhysterida, which prohibits RDP from

assigning a taxonomic label beyond order at the chosen confidence level. Consequently, the large proportion of Unassigned ASVs in our data is not solely the result of an admittedly incomplete deep-sea nematode reference database, but also of the chosen taxonomic assignment method. Nematoda, Genus-assigned and Unassigned ASVs all exhibited phylogenetic clustering, suggesting that the latter represents high congeneric richness within the genera that were successfully identified, rather than newly discovered ones. The chosen primer set will inevitably introduce a bias in our analysis, potentially over- and/or underestimating genus-level richness. Concurrently, given that the overall co-occurrence/phylogenetic patterns described herein are consistent at both phylum and genus level, and the most abundant genera found in morphological assessments of the same CCFZ samples [24] overlap with those identified in our ASVs, we believe our metabarcoding protocol is well suited to describing these deep-sea nematode assemblages.

## 5. Conclusion

Our data show that free-living deep-sea nematodes of the CCFZ are phylogenetically clustered and predominantly structured via the influence of the environment rather than intra- and interspecific interactions; these assemblages are thus more likely to be vulnerable to environmental changes, such as the perturbations brought about by large scale deep-sea mining operations. The suppressed role of competition in nematodes of the CCFZ conforms to the 'stability-time' hypothesis in which environmental stability over evolutionary timescales has enabled niche diversification in the present and thus high numbers of co-occurring taxa. ASV rarity was high throughout, represented by both low numbers of common and a substantial amount of unique ASV, highlighting the singular nature of each area. This fact calls into question the representativeness and effectiveness of the selected APEIs as source locations for preserving and replenishing biodiversity in the CCFZ. Rare taxa have been shown to carry unique functional traits [67]; concurrently the sustained provision of ecosystem services via such traits is dependent upon recolonization capacity and an absence of extinctions [18]. Sediment plumes released into the water column of the CCFZ from mining activities are expected to persist for many years and alter epi- and mesopelagic bacterioplankton communities, which are themselves structured by environmental filtering [9,76]. Interconnectedness of benthic and pelagic realms [77,78], suggests this indirect aspect of mining alone will have major consequences at the abyssal seabed.

Data accessibility. Environmental variables: Pangea, https://doi.org/10.1594/PANGAEA.873274; high-throughput sequencing data: NCBI BioProject accession no. PRJNA544999

Authors' contributions. L.M. and A.V. collected the field data, L.M. and A.R. carried out the molecular work, L.M. completed all data analysis and composed the manuscript. S.D. participated in defining the analyses performed. S.D. and A.V. participated in the design of the study and critically revised the manuscript. All authors gave final approval for publication and agree to be held accountable for the work performed therein.

Competing interests. We declare we have no competing interests.

Funding. This study was supported by Fonds Wetenschappelijk Onderzoek (FWO), grant no. 3F015515.

Acknowledgements. This work was completed within the framework of the 'Mining Impact' research project of the Joint Programming Initiative Healthy and Productive Seas and Oceans (JPI Oceans) action on

'Ecological Aspects of Deep-Sea Mining' and additionally funded by the Flanders Research Foundation (FWO, grant no. 3F015515). The molecular research was carried out with infrastructure funded by EMBRC Belgium (FWO project GOH3817N). The authors thank the captain, crew, chief scientist and all scientific staff for their invaluable help during the SO239 campaign (funding by the German Federal Ministry of Education and Research BMBF, grant no. 03F0707A-G). The assistance of Bart Beuselinck and Niels Viaene in the analysis of environmental variables is greatly appreciated. We also thank Renata Mamede da Silva Alves (alvesrms@hotmail.com) for generating figure 1.

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
