## [Reviewer comments · Proceedings of the Royal Society B: Biological Sciences]

Review History

RSPB-2019-1307.R0 (Original submission)

Review form: Reviewer 1

Recommendation

Major revision is needed (please make suggestions in comments)

Scientific importance: Is the manuscript an original and important contribution to its field?

Excellent

General interest: Is the paper of sufficient general interest?

Excellent

Quality of the paper: Is the overall quality of the paper suitable?

Good

Is the length of the paper justified?

Yes

Should the paper be seen by a specialist statistical reviewer?

No

Do you have any concerns about statistical analyses in this paper? If so, please specify them explicitly in your report.

No

It is a condition of publication that authors make their supporting data, code and materials available - either as supplementary material or hosted in an external repository. Please rate, if applicable, the supporting data on the following criteria.

Is it accessible?

No

Is it clear?

N/A

Is it adequate?

N/A

Do you have any ethical concerns with this paper?

No

Comments to the Author

This manuscript presents a metabarcoding analysis of phylogenetic patterns and species distributions of free-living marine nematodes isolated from six geographically distinct deep-sea areas in the Clarion-Clipperton Fracture Zone (CCFZ). These benthic CCFZ habitats are of critical interest for deep-sea mining projects, and thus knowledge of biodiversity and phylogeography is critical for promoting habitat preservation and preventing biodiversity loss due to anthropogenic impacts. This sample set is very unique and the study is well-designed and executed; the authors have carried out quite an extensive set of analyses assessing co-occurrence patterns across different nematode genera and geographic areas. However, I found the presentation of the results a bit difficult to follow (especially some of the visualizations in Figures 2-7 and the lack of phylogenetic trees). This manuscript could be significantly strengthened by including additional analyses and clarifying some terms/patterns presented in the introduction and discussion (see major points below). Overall, I thoroughly enjoyed reading this manuscript and believe that it significantly advances our knowledge of deep-sea biodiversity patterns.

Major Comments:

* For a manuscript that focuses on phylogenetic patterns, I was very surprised at the lack of phylogenetic tree figures (or any kind of visual display/assessment of phylogenetic clade structure of dominant nematode genera found across different CCFZ sampling areas). This type of visualization and analysis seems like a critical missing piece - I would recommend that the authors focus on tree building and phylogeny visualization for the three main dominant genera recovered in this study (*Acantholaimus*, *Halalaimus*, and *Desmoscolex*). These three genera have different putative evolutionary origins (deep-sea vs. shallow water) and life histories and ecology (predatory vs. bacterial feeding taxa). Phylogenies would be especially useful for shedding light on 1) the number of phylogenetically distinct ASVs, and the clade structure and PD between different abundant/rare ASVs attributed to each of the three nematode genera, and 2) the relationship between geography and phylogenetic clade structure - for example, do co-occurring taxa represent early-branching (basal) clades or are they more recent evolutionary radiations arising within a group? Furthermore, do any geographic areas in the CCFZ harbor unique/divergent lineages? Phylogenetic analysis can be easily accomplished for ASVs using software such as pplacer (<https://matsen.fhrc.org/pplacer/>) or the RAxML-based Evolutionary

Placement Algorithm (<https://cme.h-its.org/exelixis/web/software/epa/index.html>), in conjunction with the reference 18S sequence database the authors compiled for this study.

* Were replicate sediment cores collected in each sampling area (e.g. multiple cores analyzed from one or several multicorer deployments at a station)? The authors mention "CCFZ replicates indicated co-occurrences that were higher than expected by chance" (line 233), but Supplementary Table 1 only seems to indicate different stations in each CCFZ area. It is not clear whether the authors pooled data from replicate cores. Clarifying the availability/analysis of core replicates is important for the overarching questions being asked here - one thing the authors do not really discuss is the role of micro-habitat heterogeneity and the potential impact of micro-scale features on nematode dispersal, survival, and extinction. Since POC flux and thus food availability is patchily distributed in the deep sea, replicate cores from a single multicorer deployment could have very different environmental conditions and resource availability (in terms of TOC, bacterial/archaeal communities, predators/competitors/pathogens). Thus a "rare" species in one core could be abundant in an adjacent patch of sediment and potentially not at major risk of local extinction (only extinction within one particular core due to micro-scale conditions). Regardless of whether the authors have replicate core data, I would like to see an expanded discussion of such micro-scale patterns (and analysis of aggregation/segregation of nematode ASVs across replicates obtained from a single site/station, if the authors did indeed collect replicate cores).

* (Line 317-336 and elsewhere) I would recommend further clarification of the how the terms "environmental characteristics" and "competitive interactions" are being used in this manuscript. When the authors are talking about environmental filtering, are they only referring to abiotic conditions (e.g. TOC, sediment grain size)? Or does this also take into account biological interactions (e.g. availability/biomass of microbial food sources, pathogens such as viruses and fungi, etc.). For nematodes that co-exist, are "conserved traits" (line 321) referring to the nematode itself (mouth type, life history) or the surrounding environment (presence of prey species or microbial food source)?

* The authors also do not make use of nematode-specific ecological analyses such as feeding groups (*sensu* Wieser 1953) or maturity index (Bongers 1990) - I am curious to know whether genera with certain life histories (e.g. epistrate feeders such as *Acantholaimus*) always show the same patterns of co-occurrence in the CCFZ, or if "rare" species are most commonly associated with a specific feeding group (e.g. 1A selective deposit feeders, that may feed on a specific bacterial strain). Although this type of analysis is not as critical as phylogenetic trees (see above), adding further results and discussion of these type of patterns would significantly strengthen the conclusions presented here.

Minor Comments:

Line 93: change to "tubes were incubated for"

Line 95: change to "of sterile water was added"

Lines 118-128: DADA2 and QIIME1 are acronyms and thus should be always be capitalized

Line 120: You do not need to cite the specific URL of the DADA2 tutorial you used, referencing the software itself is enough. However, you should indicate which steps you used default parameters, and where you modified/customized any parameters because of specific features of your dataset.

Line 120: If possible, please post command line scripts and parameters on GitHub or Figshare - this will help other people replicate your analyses in the future (e.g. for DADA2, QIIME, and RStudio analyses)

Line 285: "I wonder if we can not remove..."  I think the authors forgot to remove this sentence from the text before submission

Line 347: "Rarity in the CCFZ"  incomplete sentence

Review form: Reviewer 2

Recommendation

Reject - article is not of sufficient interest (we will consider a transfer to another journal)

Scientific importance: Is the manuscript an original and important contribution to its field?

Acceptable

General interest: Is the paper of sufficient general interest?

Acceptable

Quality of the paper: Is the overall quality of the paper suitable?

Acceptable

Is the length of the paper justified?

Yes

Should the paper be seen by a specialist statistical reviewer?

Yes

Do you have any concerns about statistical analyses in this paper? If so, please specify them explicitly in your report.

Yes

It is a condition of publication that authors make their supporting data, code and materials available - either as supplementary material or hosted in an external repository. Please rate, if applicable, the supporting data on the following criteria.

Is it accessible?

N/A

Is it clear?

N/A

Is it adequate?

N/A

Do you have any ethical concerns with this paper?

No

Comments to the Author

The topic covered by this manuscript is very interesting, especially in relation to the organisms analyzed. Nonetheless, it was a bit hard to follow the analytical design, not because the analyses were complex, but rather because the sampling design is not clear enough. How many observations were analyzed in each analysis set? That was not clear to me. Some additional comments:

1) I see no reason to use DistLM in cases where the response variable is not a distance matrix. It would be justifiable for UniFrac analysis, but not to PD/MPD ones. Furthermore, it is a bit hard to follow the analyses since sampling design is not too clear. I only got the information on the number of observations in the analyses in the Results section.

2) You analyzed the association between diversity measures and environmental gradients. That is hardly enough to uncover mechanisms driving diversity patterns. Please be less ambitious with your goals.

Decision letter (RSPB-2019-1307.R0)

17-Jul-2019

Dear Miss Macheriotou:

I am writing to inform you that your manuscript RSPB-2019-1307 entitled "Environmental filtering and rarity imply high risk of species extinction in prospective deep-sea mining areas of the Clarion-Clipperton Fracture Zone" has, in its current form, been rejected for publication in Proceedings B.

This action has been taken on the advice of referees, who have recommended that substantial revisions are necessary. With this in mind we would be happy to consider a resubmission, provided the comments of the referees are fully addressed. However please note that this is not a provisional acceptance.

Sincerely,
Dr Daniel Costa
<mailto:proceedingsb@royalsociety.org>

Associate Editor

Board Member: 1

Comments to Author:

This is an interesting paper that uses meta-barcoding to investigate patterns of phylogenetic and richness of different nematode deep-sea communities located at the Clarion-Clipperton Fracture Zone (CCFZ). The study area is of critical interest for deep-sea mining projects and hence for conservation. The paper is potentially a good contribution but it lacks clarity, especially with regards to the experimental and analytical designs, which precludes its proper evaluation (see comments of both reviewer). As pointed out by reviewer 1, it is also sometimes hard to follow. For example, the methods (see reviewer 1) and results section could be tightened up in some parts and better explained in others. As it stands there are several results at different hierarchical levels and is very hard for the reader to follow (in the discussion this is better) what is the main message. There is also some lack of justification to why measure and analyze the data using different hierarchical levels (ASV vs Genera; replicates vs Area). Those conceptual justifications should be clear and come up front in the paper.

The reviewers have great suggestions, especially regarding the methods, the experimental design and the presentation of some additional data/results/figures (e.g. the phylogenetic tree). Apart from those I would also add/suggest the following points for clarification and/or discussion:

1) Towards the end of the paper (e.g. lines 398 to 404) the authors point out a potential limitation of the method by using as an example the reassessment of the absence/abundance of the genera *Monhystrella* and *Thalassomonhystera*. As far as I understand, this was motivated by an a priori expectation that those should be abundant lineages (lines 38 to 384). I wonder how pervasive is this identification limitation to other taxa, where no priori expectation exists. The authors do mention this limitation but it might be also interesting to discuss this more explicitly from the point of view of the "experiment" done here to measure species and phylogenetic diversity. Could this produce any biases? It might not, and if this were the case it would make argument and the approach stronger.

2) I might have missed something but I wonder how to reconcile the idea that competition is not relevant with the fact that most species (or better saying ASV) are unique among different sites. Could this high level of uniqueness not represent competitive exclusion? How does that speak to the analysis done to test for aggregation/separation?

3) Judging from the reviewer's comments, and my own impression, I think a cartoon showing the experimental design would help the reader better understand the methodological and analytical framework.

4) Better explain how the null models were built (e.g. lines 148-149; lines 175-177). The null models are central to the argument made here so it must be crystal clear what exactly was done. Related to that, I was not sure if I got right what exactly is the presence-absence matrix. Is each column one of the 23 sites and each row a pair of taxa?

Minor:

1) Line 149: What is "EcoSimR"? I presume an R package. I citation should be added here.

2) Lines 285 to 286: There seems to be a sentence that was "left" from the editorial work done by the authors....

3) Line 347: There is bit of a sentence left in the text...

4) Figure 3 needs a better explanation.

5) A good portion of the DNA methods (section 2.2.) could be transferred to the supplemental material.

Although the manuscript has potential there are too many issues that need to be fixed, including a better description of the experimental and analytical designs to allow a proper evaluation of the paper. Given the journal policy of not allowing for multiple rounds of review, and the several issues raised, if it is in the author's interest to resubmit a new version, I suggest the authors to address all comments before considering a resubmission.

Reviewer(s)' Comments to Author:

Referee: 1

Comments to the Author(s)

This manuscript presents a metabarcoding analysis of phylogenetic patterns and species distributions of free-living marine nematodes isolated from six geographically distinct deep-sea areas in the Clarion-Clipperton Fracture Zone (CCFZ). These benthic CCFZ habitats are of critical interest for deep-sea mining projects, and thus knowledge of biodiversity and phylogeography is critical for promoting habitat preservation and preventing biodiversity loss due to anthropogenic impacts. This sample set is very unique and the study is well-designed and executed; the authors have carried out quite an extensive set of analyses assessing co-occurrence patterns across different nematode genera and geographic areas. However, I found the presentation of the results a bit difficult to follow (especially some of the visualizations in Figures 2-7 and the lack of phylogenetic trees). This manuscript could be significantly strengthened by including additional analyses and clarifying some terms/patterns presented in the introduction and discussion (see major points below). Overall, I thoroughly enjoyed reading this manuscript and believe that it significantly advances our knowledge of deep-sea biodiversity patterns.

Major Comments:

* For a manuscript that focuses on phylogenetic patterns, I was very surprised at the lack of phylogenetic tree figures (or any kind of visual display/assessment of phylogenetic clade structure of dominant nematode genera found across different CCFZ sampling areas). This type of visualization and analysis seems like a critical missing piece - I would recommend that the authors focus on tree building and phylogeny visualization for the three main dominant genera recovered in this study (*Acantholaimus*, *Halalaimus*, and *Desmoscolex*). These three genera have different putative evolutionary origins (deep-sea vs. shallow water) and life histories and ecology (predatory vs. bacterial feeding taxa). Phylogenies would be especially useful for shedding light on 1) the number of phylogenetically distinct ASVs, and the clade structure and PD between different abundant/rare ASVs attributed to each of the three nematode genera, and 2) the relationship between geography and phylogenetic clade structure - for example, do co-occurring taxa represent early-branching (basal) clades or are they more recent evolutionary radiations arising within a group? Furthermore, do any geographic areas in the CCFZ harbor unique/divergent lineages? Phylogenetic analysis can be easily accomplished for ASVs using software such as pplacer (<https://matsen.fhrc.org/pplacer/>) or the RAxML-based Evolutionary Placement Algorithm (<https://cme.h-its.org/exelixis/web/software/epa/index.html>), in conjunction with the reference 18S sequence database the authors compiled for this study.

* Were replicate sediment cores collected in each sampling area (e.g. multiple cores analyzed from one or several multicorer deployments at a station)? The authors mention "CCFZ replicates indicated co-occurrences that were higher than expected by chance" (line 233), but Supplementary Table 1 only seems to indicate different stations in each CCFZ area. It is not clear whether the authors pooled data from replicate cores. Clarifying the availability/analysis of core replicates is important for the overarching questions being asked here - one thing the authors do

not really discuss is the role of micro-habitat heterogeneity and the potential impact of micro-scale features on nematode dispersal, survival, and extinction. Since POC flux and thus food availability is patchily distributed in the deep sea, replicate cores from a single multicorer deployment could have very different environmental conditions and resource availability (in terms of TOC, bacterial/archaeal communities, predators/competitors/pathogens). Thus a "rare" species in one core could be abundant in an adjacent patch of sediment and potentially not at major risk of local extinction (only extinction within one particular core due to micro-scale conditions). Regardless of whether the authors have replicate core data, I would like to see an expanded discussion of such micro-scale patterns (and analysis of aggregation/segregation of nematode ASVs across replicates obtained from a single site/station, if the authors did indeed collect replicate cores).

* (Line 317-336 and elsewhere) I would recommend further clarification of the how the terms "environmental characteristics" and "competitive interactions" are being used in this manuscript. When the authors are talking about environmental filtering, are they only referring to abiotic conditions (e.g. TOC, sediment grain size)? Or does this also take into account biological interactions (e.g. availability/biomass of microbial food sources, pathogens such as viruses and fungi, etc.). For nematodes that co-exist, are "conserved traits" (line 321) referring to the nematode itself (mouth type, life history) or the surrounding environment (presence of prey species or microbial food source)?

* The authors also do not make use of nematode-specific ecological analyses such as feeding groups (*sensu* Wieser 1953) or maturity index (Bongers 1990) - I am curious to know whether genera with certain life histories (e.g. epistrate feeders such as *Acantholaimus*) always show the same patterns of co-occurrence in the CCFZ, or if "rare" species are most commonly associated with a specific feeding group (e.g. 1A selective deposit feeders, that may feed on a specific bacterial strain). Although this type of analysis is not as critical as phylogenetic trees (see above), adding further results and discussion of these type of patterns would significantly strengthen the conclusions presented here.

Minor Comments:

Line 93: change to "tubes were incubated for"

Line 95: change to "of sterile water was added"

Lines 118-128: DADA2 and QIIME1 are acronyms and thus should be always be capitalized

Line 120: You do not need to cite the specific URL of the DADA2 tutorial you used, referencing the software itself is enough. However, you should indicate which steps you used default parameters, and where you modified/customized any parameters because of specific features of your dataset.

Line 120: If possible, please post command line scripts and parameters on GitHub or Figshare - this will help other people replicate your analyses in the future (e.g. for DADA2, QIIME, and RStudio analyses)

Line 285: "I wonder if we can not remove..."  I think the authors forgot to remove this sentence from the text before submission

Line 347: "Rarity in the CCFZ"  incomplete sentence

Referee: 2

Comments to the Author(s)

The topic covered by this manuscript is very interesting, especially in relation to the organisms analyzed. Nonetheless, it was a bit hard to follow the analytical design, not because the analyses were complex, but rather because the sampling design is not clear enough. How many observations were analyzed in each analysis set? That was not clear to me. Some additional comments:

1) I see no reason to use DistLM in cases where the response variable is not a distance matrix. It would be justifiable for UniFrac analysis, but not to PD/MPD ones. Furthermore, it is a bit hard to follow the analyses since sampling design is not too clear. I only got the information on the number of observations in the analyses in the Results section.

2) You analyzed the association between diversity measures and environmental gradients. That is hardly enough to uncover mechanisms driving diversity patterns. Please be less ambitious with your goals.

Author's Response to Decision Letter for (RSPB-2019-1307.R0)

See Appendix A.

RSPB-2019-2666.R0

Review form: Reviewer 1

Recommendation

Accept with minor revision (please list in comments)

Scientific importance: Is the manuscript an original and important contribution to its field?

Excellent

General interest: Is the paper of sufficient general interest?

Excellent

Quality of the paper: Is the overall quality of the paper suitable?

Excellent

Is the length of the paper justified?

Yes

Should the paper be seen by a specialist statistical reviewer?

No

Do you have any concerns about statistical analyses in this paper? If so, please specify them explicitly in your report.

No

It is a condition of publication that authors make their supporting data, code and materials available - either as supplementary material or hosted in an external repository. Please rate, if applicable, the supporting data on the following criteria.

Is it accessible?

Yes

Is it clear?

Yes

Is it adequate?

Yes

Do you have any ethical concerns with this paper?

No

Comments to the Author

The authors have done an excellent job at reworking the original manuscript, and the narrative is clearer and more concise in this resubmission. The discussion is now much stronger and the take-home points are clearly conveyed; I believe the results here are a very important scientific contribution to the literature focused on deep-sea mining impacts. I also appreciate the authors' efforts to incorporate more discussion of phylogenetic results and patterns (taking note of their intention to explore these aspects more fully in a future manuscript) - my original concerns and comments have been fully addressed, I had only one more minor comment in regard to a supplementary figure, as follows:

Figure S4 - The layout of this figure is confusing - could you clarify which bar charts represent the shared and unique ASVs (I could not figure this out by looking at the axis labels or caption, but it seems like there are some important patterns here).

Review form: Reviewer 2

Recommendation

Reject - article is not of sufficient interest (we will consider a transfer to another journal)

Scientific importance: Is the manuscript an original and important contribution to its field?

Acceptable

General interest: Is the paper of sufficient general interest?

Acceptable

Quality of the paper: Is the overall quality of the paper suitable?

Acceptable

Is the length of the paper justified?

Yes

Should the paper be seen by a specialist statistical reviewer?

Yes

Do you have any concerns about statistical analyses in this paper? If so, please specify them explicitly in your report.

Yes

It is a condition of publication that authors make their supporting data, code and materials available - either as supplementary material or hosted in an external repository. Please rate, if applicable, the supporting data on the following criteria.

Is it accessible?

Yes

Is it clear?

Yes

Is it adequate?

Yes

Do you have any ethical concerns with this paper?

No

Comments to the Author

My previous comments were considered in relation sampling design and goals. I still think there is no reason to use a distance-based analysis where the variable is not a distance matrix. I agree that is not wrong, but it is for sure unnecessary. That said, I have nothing more to add in this review.

Review form: Reviewer 3 (Oliver S. Ashford)

Recommendation

Accept with minor revision (please list in comments)

Scientific importance: Is the manuscript an original and important contribution to its field?

Excellent

General interest: Is the paper of sufficient general interest?

Excellent

Quality of the paper: Is the overall quality of the paper suitable?

Good

Is the length of the paper justified?

Yes

Should the paper be seen by a specialist statistical reviewer?

No

Do you have any concerns about statistical analyses in this paper? If so, please specify them explicitly in your report.

Yes

It is a condition of publication that authors make their supporting data, code and materials available - either as supplementary material or hosted in an external repository. Please rate, if applicable, the supporting data on the following criteria.

Is it accessible?

Yes

Is it clear?

Yes

Is it adequate?

Yes

Do you have any ethical concerns with this paper?

No

Comments to the Author

Review of Macheriotou et al. 2019

'Phylogenetic clustering and rarity imply high risk of local species extinction in prospective deep-sea mining areas of the Clarion-Clipperton Zone'

Proceedings of the Royal Society B: Biological Sciences

Macheriotou et al. investigate the prevalence of rare taxa and phylogenetic diversity of nematode assemblages in the Clarion Clipperton Fracture Zone (CCFZ). Investigation of phylogenetic diversity has only rarely been attempted in deep-sea environments, yet holds great promise for elucidating the relative importance of different factors in shaping organismal communities. Further, the CCFZ is a zone of intense scientific and industrial interest at present because of its juxtaposition of seemingly high environmental sensitivity with high commercial value. Because of this, I was excited to review this manuscript, and I believe it would have great appeal to the broad audience of Proceedings B.

The analyses of Macheriotou et al. are generally well designed and implemented. They demonstrate a prevalence of rare taxa, and suggest assemblages to be typically phylogenetically underdispersed. These results imply a high sensitivity of CCFZ nematode assemblages to environmental perturbation, and a high risk of local species extinction should assemblages be disturbed by mining activities.

I was asked to review this paper following an initial round of review, and I am generally satisfied with the authors' response to prior review comments. However, I would like to see the authors address a small number of points prior to manuscript acceptance.

In particular, I would like the authors to defend their use of DISTLM in the manuscript when the use of a generalised linear or additive model might be considered more appropriate. I would like the authors to explain why more environmental data was not available for the study (e.g. temperature, oxygen concentration, current speed, bathymetric position index, seafloor slope), and I would like a correlation matrix of the environmental data analysed to be added to the ESM.

I was confused by the lack of an abstract, and I suggest the authors consider adding mention of the relationship between degree of phylogenetic dispersion and sensitivity of communities to environmental change to their introduction and discussion sections.

Please see below for more specific comments.

Title

Suggest changing to 'Clarion-Clipperton Fracture Zone' to match the rest of the manuscript.

Abstract

I cannot find an abstract in the manuscript. This this intentional? Please add an abstract.

Introduction

L12: Change comma to semicolon.

L14: Suggest listing some reasons for increasing demand for these metals. E.g. electric car batteries.

L18: Change to '(~4-5 km depth)'.

L31-32: Suggest changing to 'These deviations from randomness are thought to be the result of species interactions, habitat preferences, and/or speciation processes'.

L35-46: You might consider adding a few sentences outlining how communities structured by abiotic factors are generally more sensitive to changes in those factors than are communities structured predominantly by biotic factors.

L63: Write out ASV in full here.

Materials and methods

Please explain why more environmental information is not available – e.g. temperature, oxygen concentration, current speed, bathymetric position index and seafloor slope may be important factors in shaping deep-sea communities.

L80-82: Clearly too late for this now, but I would like to have seen environmental data and biological data originating from the same core by subsampling.

L109: Please state that 'UpSetR' is an R package. Same for 'EcoSimR' at line 123, and 'DECIPHER' at line 135.

L141: Change to 'Phylogenetic Diversity (PD – see below)...'.

L164-170: As mentioned above, I believe that a Generalised Linear/Additive Model would be a more appropriate analysis of relationships between PD and the environmental variables. I would also be interested to see a correlation matrix for the environmental variables. Cases of high multicollinearity can compromise statistical analyses for these types of models.

Results

L184: I assume these are mean values and standard errors? If so, please state this explicitly.

L186-187: Change to '...genera than IOM.C, BGR.RA and BGR.PA, while IFREMER had significantly fewer genera than BGR.RA and BGR.PA...'.

L187: Why is 'Area' capitalised throughout the manuscript?

L199-203: Would it be possible to put directionality on these results? E.g. 'PD increased with increasing %TOC'.

L217: '...in 3-4 out of six Areas'. Do the authors mean three or four?

Discussion

I was surprised not to find any explicit comparison of your results with those of the few other studies which have investigated phylogenetic structure in the deep sea. I suggest adding a few sentences relating to this.

L280: 'In the absence of competition...'. Make this the start of a new paragraph.

L290: 'Random co-occurrence patterns...'. Make this the start of a new paragraph.

L297: 'An additional striking difference...'. Make this the start of a new paragraph.

L299: '...extreme conditions'. This is a subjective term. Extreme for humans, yes, but not necessary extreme for other forms of life.

L306: Change to '...the presence of common genera'.

L309: 85% by taxa, or by percent of total reads?

L338-339: 'It seems likely...'. I'm not sure I agree with your view here. Small APEIs within mining areas may be heavily impacted indirectly by mining activities. I think discussion of this is beyond the scope of the manuscript and should be removed.

L380-382: This is a strange sentence with which to start your conclusion. This is certainly not the primary conclusion of your paper. Instead, I think your key conclusion is that the nematode assemblages investigated were phylogenetically clustered. In turn, this clustering suggests a high sensitivity to environmental change. You need to stress this point more.

Decision letter (RSPB-2019-2666.R0)

21-Jan-2020

Dear Miss Macheriotou:

Your manuscript has now been peer reviewed and the reviews have been assessed by an Associate Editor. The reviewers' comments (not including confidential comments to the Editor) and the comments from the Associate Editor are included at the end of this email for your reference. As you will see, the reviewers and the Editors have raised some concerns with your manuscript and we would like to invite you to revise your manuscript to address them.

Research ethics:

Use of animals and field studies:

It is a condition of publication that you make available the data and research materials supporting the results in the article. Datasets should be deposited in an appropriate publicly available repository and details of the associated accession number, link or DOI to the datasets must be included in the Data Accessibility section of the article

(<https://royalsociety.org/journals/ethics-policies/data-sharing-mining/>). Reference(s) to datasets should also be included in the reference list of the article with DOIs (where available).

If you wish to submit your data to Dryad (<http://datadryad.org/>) and have not already done so you can submit your data via this link [http://datadryad.org/submit?journalID=RSPB&manu=\(Document not available\)](http://datadryad.org/submit?journalID=RSPB&manu=(Document%20not%20available)), which will take you to your unique entry in the Dryad repository.

Please submit a copy of your revised paper within three weeks. If we do not hear from you within this time your manuscript will be rejected. If you are unable to meet this deadline please let us know as soon as possible, as we may be able to grant a short extension.

Best wishes,
Dr Daniel Costa
mailto: proceedingsb@royalsociety.org

Reviewer(s)' Comments to Author:
Referee: 3

Comments to the Author(s).

Review of Macheriotou et al. 2019

'Phylogenetic clustering and rarity imply high risk of local species extinction in prospective deep-sea mining areas of the Clarion-Clipperton Zone'

Proceedings of the Royal Society B: Biological Sciences

Macheriotou et al. investigate the prevalence of rare taxa and phylogenetic diversity of nematode assemblages in the Clarion Clipperton Fracture Zone (CCFZ). Investigation of phylogenetic diversity has only rarely been attempted in deep-sea environments, yet holds great promise for elucidating the relative importance of different factors in shaping organismal communities. Further, the CCFZ is a zone of intense scientific and industrial interest at present because of its

juxtaposition of seemingly high environmental sensitivity with high commercial value. Because of this, I was excited to review this manuscript, and I believe it would have great appeal to the broad audience of Proceedings B.

The analyses of Macheriotou et al. are generally well designed and implemented. They demonstrate a prevalence of rare taxa, and suggest assemblages to be typically phylogenetically underdispersed. These results imply a high sensitivity of CCFZ nematode assemblages to environmental perturbation, and a high risk of local species extinction should assemblages be disturbed by mining activities.

I was asked to review this paper following an initial round of review, and I am generally satisfied with the authors' response to prior review comments. However, I would like to see the authors address a small number of points prior to manuscript acceptance.

In particular, I would like the authors to defend their use of DISTLM in the manuscript when the use of a generalised linear or additive model might be considered more appropriate. I would like the authors to explain why more environmental data was not available for the study (e.g. temperature, oxygen concentration, current speed, bathymetric position index, seafloor slope), and I would like a correlation matrix of the environmental data analysed to be added to the ESM.

I was confused by the lack of an abstract, and I suggest the authors consider adding mention of the relationship between degree of phylogenetic dispersion and sensitivity of communities to environmental change to their introduction and discussion sections. Please see below for more specific comments.

Title

Suggest changing to 'Clarion-Clipperton Fracture Zone' to match the rest of the manuscript.

Abstract

I cannot find an abstract in the manuscript. This this intentional? Please add an abstract.

Introduction

L12: Change comma to semicolon.

L14: Suggest listing some reasons for increasing demand for these metals. E.g. electric car batteries.

L18: Change to '(~4-5 km depth)'.

L31-32: Suggest changing to 'These deviations from randomness are thought to be the result of species interactions, habitat preferences, and/or speciation processes'.

L35-46: You might consider adding a few sentences outlining how communities structured by abiotic factors are generally more sensitive to changes in those factors than are communities structured predominantly by biotic factors.

L63: Write out ASV in full here.

Materials and methods

Please explain why more environmental information is not available - e.g. temperature, oxygen concentration, current speed, bathymetric position index and seafloor slope may be important factors in shaping deep-sea communities.

L80-82: Clearly too late for this now, but I would like to have seen environmental data and biological data originating from the same core by subsampling.

L109: Please state that 'UpSetR' is an R package. Same for 'EcoSimR' at line 123, and 'DECIPHER' at line 135.

L141: Change to 'Phylogenetic Diversity (PD - see below)...'.

L164-170: As mentioned above, I believe that a Generalised Linear/Additive Model would be a more appropriate analysis of relationships between PD and the environmental variables. I would also be interested to see a correlation matrix for the environmental variables. Cases of high multicollinearity can compromise statistical analyses for these types of models.

Results

L184: I assume these are mean values and standard errors? If so, please state this explicitly.

L186-187: Change to '...genera than IOM.C, BGR.RA and BGR.PA, while IFREMER had significantly fewer genera than BGR.RA and BGR.PA...'

L187: Why is 'Area' capitalised throughout the manuscript?

L199-203: Would it be possible to put directionality on these results? E.g. 'PD increased with increasing %TOC'.

L217: '...in 3-4 out of six Areas'. Do the authors mean three or four?

Discussion

I was surprised not to find any explicit comparison of your results with those of the few other studies which have investigated phylogenetic structure in the deep sea. I suggest adding a few sentences relating to this.

L280: 'In the absence of competition...'. Make this the start of a new paragraph.

L290: 'Random co-occurrence patterns...'. Make this the start of a new paragraph.

L297: 'An additional striking difference...'. Make this the start of a new paragraph.

L299: '...extreme conditions'. This is a subjective term. Extreme for humans, yes, but not necessary extreme for other forms of life.

L306: Change to '...the presence of common genera'.

L309: 85% by taxa, or by percent of total reads?

L338-339: 'It seems likely...'. I'm not sure I agree with your view here. Small APEIs within mining areas may be heavily impacted indirectly by mining activities. I think discussion of this is beyond the scope of the manuscript and should be removed.

L380-382: This is a strange sentence with which to start your conclusion. This is certainly not the primary conclusion of your paper. Instead, I think your key conclusion is that the nematode assemblages investigated were phylogenetically clustered. In turn, this clustering suggests a high sensitivity to environmental change. You need to stress this point more.

Referee: 2

Comments to the Author(s).

My previous comments were considered in relation sampling design and goals. I still think there is no reason to use a distance-based analysis where the variable is not a distance matrix. I agree that is not wrong, but it is for sure unnecessary. That said, I have nothing more to add in this review.

Referee: 1

Comments to the Author(s).

The authors have done an excellent job at reworking the original manuscript, and the narrative is clearer and more concise in this resubmission. The discussion is now much stronger and the take-home points are clearly conveyed; I believe the results here are a very important scientific contribution to the literature focused on deep-sea mining impacts. I also appreciate the authors' efforts to incorporate more discussion of phylogenetic results and patterns (taking note of their intention to explore these aspects more fully in a future manuscript) - my original concerns and comments have been fully addressed, I had only one more minor comment in regard to a supplementary figure, as follows:

Figure S4 - The layout of this figure is confusing - could you clarify which bar charts represent the shared and unique ASVs (I could not figure this out by looking at the axis labels or caption, but it seems like there are some important patterns here).

Author's Response to Decision Letter for (RSPB-2019-2666.R0)

See Appendix B.

RSPB-2019-2666.R1 (Revision)

Review form: Reviewer 3 (Oliver S. Ashford)

Recommendation

Accept as is

Scientific importance: Is the manuscript an original and important contribution to its field?

Excellent

General interest: Is the paper of sufficient general interest?

Excellent

Quality of the paper: Is the overall quality of the paper suitable?

Good

Is the length of the paper justified?

Yes

Should the paper be seen by a specialist statistical reviewer?

No

Do you have any concerns about statistical analyses in this paper? If so, please specify them explicitly in your report.

No

It is a condition of publication that authors make their supporting data, code and materials available - either as supplementary material or hosted in an external repository. Please rate, if applicable, the supporting data on the following criteria.

Is it accessible?

Yes

Is it clear?

Yes

Is it adequate?

Yes

Do you have any ethical concerns with this paper?

No

Comments to the Author

Review of Macheriotou et al. 2020

RSPB-2019-2666.R1

'Phylogenetic clustering and rarity imply risk of local species extinction in prospective deep-sea

mining areas of the Clarion-Clipperton Fracture Zone'
 Proceedings of the Royal Society B: Biological Sciences

I'd like to thank the authors for their work revising the manuscript. I feel that all my previous recommendations have been given fair consideration, and that the manuscript has been improved.

I now look very positively on the objectives of this manuscript and its analysis methodology, and believe that its findings will have great appeal to the broad audience of Proceedings B.

I have just two very minor comments, which the authors may feel free to ignore:

L85-86: Suggest changing '...a 1mL was subsampled from...' to '...a 1 mL subsample was taken from...'

L137: It might be clearer to capitalise 'standardized effect size' - ses throughout the manuscript text.

Decision letter (RSPB-2019-2666.R1)

12-Mar-2020

Dear Miss Macheriotou

I am pleased to inform you that your Review manuscript RSPB-2019-2666.R1 entitled "Phylogenetic clustering and rarity imply risk of local species extinction in prospective deep-sea mining areas of the Clarion-Clipperton Fracture Zone" has been accepted for publication in Proceedings B.

The referee(s) do not recommend any further changes. Therefore, please proof-read your manuscript carefully and upload your final files for publication. Because the schedule for publication is very tight, it is a condition of publication that you submit the revised version of your manuscript within 7 days. If you do not think you will be able to meet this date please let me know immediately.

To upload your manuscript, log into <http://mc.manuscriptcentral.com/prsb> and enter your Author Centre, where you will find your manuscript title listed under "Manuscripts with Decisions." Under "Actions," click on "Create a Revision." Your manuscript number has been appended to denote a revision.

You will be unable to make your revisions on the originally submitted version of the manuscript. Instead, upload a new version through your Author Centre.

- 1) A text file of the manuscript (doc, txt, rtf or tex), including the references, tables (including captions) and figure captions. Please remove any tracked changes from the text before submission. PDF files are not an accepted format for the "Main Document".
- 2) A separate electronic file of each figure (tiff, EPS or print-quality PDF preferred). The format should be produced directly from original creation package, or original software format. Please note that PowerPoint files are not accepted.
- 3) Electronic supplementary material: this should be contained in a separate file from the main

text and the file name should contain the author's name and journal name, e.g
 authorname_procb_ESM_figures.pdf

All supplementary materials accompanying an accepted article will be treated as in their final form. They will be published alongside the paper on the journal website and posted on the online figshare repository. Files on figshare will be made available approximately one week before the accompanying article so that the supplementary material can be attributed a unique DOI. Please see: <https://royalsociety.org/journals/authors/author-guidelines/>

4) Data-Sharing and data citation

It is a condition of publication that data supporting your paper are made available. Data should be made available either in the electronic supplementary material or through an appropriate repository. Details of how to access data should be included in your paper. Please see <https://royalsociety.org/journals/ethics-policies/data-sharing-mining/> for more details.

<http://datadryad.org/submit?journalID=RSPB&manu=RSPB-2019-2666.R1> which will take you to your unique entry in the Dryad repository.

Once again, thank you for submitting your manuscript to Proceedings B and I look forward to receiving your final version. If you have any questions at all, please do not hesitate to get in touch.

Sincerely,

Dr Daniel Costa

Associate Editor Board Member: 1

Comments to Author:

This is a much-improved version of the manuscript and I agree with the reviewer's (last round one review only, but I am taking the whole history here) that the paper will be a nice and relevant contribution. I think the paper is basically ready for publication and apart from the minor suggestions given by the last reviewer, there are a few minor aspects that I think need to be taken care of before the paper is finally accepted for publication. Of particular importance are:

1- I could not find the figure captions for most figures except figure 1. Although this was not crucial on my full evaluation of the paper's merit, it did make my evaluation a bit harder and it would be nice to be able to evaluate it before we have a final version.

2- In some figures, and this is particularly problematic because there were no captions, the y-axis simply says "value" (e.g. Figure 4 where the "value" I think represents the phylogenetic diversity metric). Without a caption this becomes even less informative, so I would recommend the addition, when possible, of a bit more info of what the y-axis is.

Reviewer(s)' Comments to Author:

Referee: 3

Comments to the Author(s)

Review of Macheriotou et al. 2020

RSPB-2019-2666.R1

'Phylogenetic clustering and rarity imply risk of local species extinction in prospective deep-sea mining areas of the Clarion-Clipperton Fracture Zone'
 Proceedings of the Royal Society B: Biological Sciences

I'd like to thank the authors for their work revising the manuscript. I feel that all my previous recommendations have been given fair consideration, and that the manuscript has been improved.

I now look very positively on the objectives of this manuscript and its analysis methodology, and believe that its findings will have great appeal to the broad audience of Proceedings B.

I have just two very minor comments, which the authors may feel free to ignore:

L85-86: Suggest changing '...a 1mL was subsampled from...' to '...a 1 mL subsample was taken from...'

L137: It might be clearer to capitalise 'standardized effect size' - ses throughout the manuscript text.

Decision letter (RSPB-2019-2666.R2)

12-Mar-2020

Dear Miss Macheriotou

I am pleased to inform you that your manuscript entitled "Phylogenetic clustering and rarity imply risk of local species extinction in prospective deep-sea mining areas of the Clarion-Clipperton Fracture Zone" has been accepted for publication in Proceedings B.

Open Access

Paper charges

Sincerely,
Editor, Proceedings B
<mailto:proceedingsb@royalsociety.org>

Appendix A

Associate Editor

Board Member: 1

Comments to Author:

This is an interesting paper that uses meta-barcoding to investigate patterns of phylogenetic and richness of different nematode deep-sea communities located at the Clarion-Clipperton Fracture Zone (CCFZ). The study area is of critical interest for deep-sea mining projects and hence for conservation. The paper is potentially a good contribution but it lacks clarity, especially with regards to the experimental and analytical designs, which precludes its proper evaluation (see comments of both reviewer). As pointed out by reviewer 1, it is also sometimes hard to follow. For example, the methods (see reviewer 1) and results section could be tightened up in some parts and better explained in others. As it stands there are several results at different hierarchical levels and is very hard for the reader to follow (in the discussion this is better) what is the main message. There is also some lack of justification to why measure and analyze the data using different hierarchical levels (ASV vs Genera; replicates vs Area). Those conceptual justifications should be clear and come up front in the paper.

Reply: The authors would like to thank the Reviewers as well as the Associate Editor for their instructive comments and suggestions. We have followed these to a major extent and modified the text, figures and ESM in order to improve clarity and content. We hope you agree and would reconsider the publication of our work in Proceedings of the Royal Society B.

We have addressed shortcomings in clarity and justification of the levels of data analysis in Lines 101-103, 113-116, 138-140.

The reviewers have great suggestions, especially regarding the methods, the experimental design and the presentation of some additional data/results/figures (e.g. the phylogenetic tree). Apart from those I would also add/suggest the following points for clarification and/or discussion:

1) Towards the end of the paper (e.g. Lines 398 to 404) the authors point out a potential limitation of the method by using as an example the reassessment of the absence/abundance of the genera *Monhystrella* and *Thalassomonhystera*. As far as I understand, this was motivated by an a priori expectation that those should be abundant Lineages (Lines 38 to 384). I wonder how pervasive is this identification limitation to other taxa, where no priori expectation exists. The authors do mention this limitation but it might be also interesting to discuss this more explicitly from the point of view of the "experiment" done here to measure species and phylogenetic diversity. Could this produce any biases? It might not, and if this were the case it would make argument and the approach stronger.

Reply: Phylogenetic diversity would be unaffected by such biases as this is based on the nucleotide sequence rather than the taxonomic label it receives. We have added a short discussion of what we expect the effects of identification would be at Lines 370-375.

2) I might have missed something but I wonder how to reconcile the idea that competition is not relevant with the fact that most species (or better saying ASV) are unique among different sites. Could this high level of uniqueness not represent competitive exclusion? How does that speak to the analysis done to test for aggregation/separation?

Reply: This high uniqueness need not be the result solely of competitive exclusion as the same patterns could arise from differential habitat preferences between taxa and/or historical/phylogenetic processes such as allopatric speciation (Horner-Devine et al., 2007). Moreover, although this pattern can indeed seem counterintuitive, i.e. the large fraction of unique ASVs and assemblages characterised by higher co-occurrences than expected by chance (aggregation). However, considering how CPs and C-score are calculated one can see how uniqueness and aggregation can co-occur. The unique ASVs comprise ~50% of the assemblage in each Area, and because they are numerous and (by definition) found in just one Area, they do not contribute significantly to the number of CPs. Consider the checkerboard pair example below; assuming both ASV 1 and 2 are unique, there will only be one pairwise combination that will result in a CP for these taxa. A larger contribution of CPs would result from an ASV that is fairly common, yet excluded from some Areas due to the presence of another ASV. Thus, due to the definition of CPs, the uniqueness of ASVs is not in fact interpreted on the basis of competitive exclusion, as manifested by CPs.

ASV	Area	
	APEI3	IFREMER
1	1	0
2	0	1

3) Judging from the reviewer's comments, and my own impression, I think a cartoon showing the experimental design would help the reader better understand the methodological and analytical framework.

Reply: We have included a diagram (Fig. 1S) of our experimental design in the ESM document in order to improve clarity.

4) Better explain how the null models were built (e.g. Lines 148-149; Lines 175-177). The null models are central to the argument made here so it must be crystal clear what exactly was done. Related to that, I was not sure if I got right what exactly is the presence-absence matrix. Is each column one of the 23 sites and each row a pair of taxa?

Reply: Co-occurrence metrics: the program processes the entire presence-absence matrix (sites as columns and taxa as rows), and quantifies the number of checkerboard pairs by analysing each pairwise combination of ASVs. Added detailed description of input data at Lines 118-121, added descriptions of the null models at Lines 123-124 (co-occurrence analysis) and ESM Table 12 (phylogenetic community structure analysis).

Minor:

1) Line 149: What is "EcoSimR"? I presume an R package. I citation should be added here.

Reply: Correct presumption, citation added at Line 123.

2) Lines 285 to 286: There seems to be a sentence that was "left" from the editorial work done by the authors....

Reply: Indeed, this slipped through the proofreading somehow, our apologies.

3) Line 347: There is bit of a sentence left in the text...

Reply: Indeed, this slipped through the proofreading somehow, our apologies.

4) Figure 3 needs a better explanation.

Reply: Modified caption for clarity.

5) A good portion of the DNA methods (section 2.2.) could be transferred to the supplemental material.

Reply: Section 2.2 Moved to ESM.

Although the manuscript has potential there are too many issues that need to be fixed, including a better description of the experimental and analytical designs to allow a proper evaluation of the paper. Given the journal policy of not allowing for multiple rounds of review, and the several issues raised, if it is in the author's interest to resubmit a new version, I suggest the authors to address all comments before considering a resubmission.

Reviewer(s)' Comments to Author:

Referee: 1

Comments to the Author(s)

This manuscript presents a metabarcoding analysis of phylogenetic patterns and species distributions of free-living marine nematodes isolated from six geographically distinct deep-sea areas in the Clarion-Clipperton Fracture Zone (CCFZ). These benthic CCFZ habitats are of critical interest for deep-sea mining projects, and thus knowledge of biodiversity and phylogeography is critical for promoting habitat preservation and preventing biodiversity loss due to anthropogenic impacts. This sample set is very unique and the study is well-designed and executed; the authors have carried out quite an extensive set of analyses assessing co-occurrence patterns across different nematode genera and geographic areas. However, I found the presentation of the results a bit difficult to follow (especially some of the visualizations in Figures 2-7 and the lack of phylogenetic trees). This manuscript could be significantly strengthened by including additional analyses and clarifying some terms/patterns presented in the introduction and discussion (see major points below). Overall, I thoroughly enjoyed reading this manuscript and believe that it significantly advances our knowledge of deep-sea biodiversity patterns.

Major Comments:

* For a manuscript that focuses on phylogenetic patterns, I was very surprised at the lack of phylogenetic tree figures (or any kind of visual display/assessment of phylogenetic clade structure of dominant nematode genera found across different CCFZ sampling areas). This type of visualization and analysis seems like a critical missing piece - I would recommend that the authors focus on tree building and phylogeny visualization for the three main dominant genera recovered in this study (*Acantholaimus*, *Halalaimus*, and *Desmoscolex*). These three genera have different putative evolutionary origins (deep-sea vs. shallow water) and life histories and ecology (predatory vs. bacterial feeding taxa). Phylogenies would be especially useful for shedding light on 1) the number of phylogenetically distinct ASVs, and the clade structure and PD between different abundant/rare ASVs attributed to each of the three nematode genera, and 2) the relationship between geography and phylogenetic clade structure - for example, do co-occurring taxa represent early-branching (basal) clades or are they more recent evolutionary radiations arising within a group? Furthermore, do any geographic areas in the CCFZ harbor unique/divergent Lineages? Phylogenetic analysis can be easily accomplished for ASVs using software such as pplacer (<https://matsen.fhcrc.org/pplacer/>) or the RAxML-based Evolutionary Placement Algorithm (<https://cme.h-its.org/exelixis/web/software/epa/index.html>), in conjunction with the reference 18S sequence database the authors compiled for this study.

Reply: We agree with the reviewer that one would reasonably expect to see phylogenetic trees as part of this study, indeed, their branch lengths are what the phylogenetic metrics (ses.PD, ses.MNTD, ses.MPD) are based upon. The requested trees are included at the end of this document. The ASVs show no clustering by Area, nor the presence of phylogenetically divergent Lineages, as would be expected given the observed absence phylogenetic overdispersion in the three dominant genera. The perspectives described by the reviewer are precisely those intended to be implemented and discussed more thoroughly in an upcoming publication, as such we chose (and hope you can agree) to exclude these from the current one.

* Were replicate sediment cores collected in each sampling area (e.g. multiple cores analyzed from one or several multicorer deployments at a station)? The authors mention "CCFZ replicates indicated co-occurrences that were higher than expected by chance" (Line 233), but Supplementary Table 1 only seems to indicate different stations in each CCFZ area. It is not clear whether the authors pooled data from replicate cores. Clarifying the availability/analysis of core replicates is important for the overarching questions being asked here - one thing the authors do not really discuss is the role of micro-habitat heterogeneity and the potential impact of micro-scale features on nematode dispersal, survival, and extinction. Since POC flux and thus food availability is patchily distributed in the deep sea, replicate cores from a single multicorer deployment could have very different environmental conditions and resource availability (in terms of TOC, bacterial/archaeal communities, predators/competitors/pathogens). Thus a "rare" species in one core could be abundant in an adjacent patch of sediment and potentially not at major risk of local extinction (only extinction within one particular core due to micro-scale conditions). Regardless of whether the authors have replicate core data, I would like to see an expanded discussion of such micro-scale patterns (and analysis of aggregation/segregation of nematode ASVs across replicates obtained from a single site/station, if the authors did indeed collect replicate cores).

Reply: One replicate core was taken from each multicore deployment, added improved description of sampling at Lines 75-77 and of the levels of co-occurrence analysis at Lines 113-121.

Added discussion of microhabitat heterogeneity at Line 239-241, 306-311.

* (Line 317-336 and elsewhere) I would recommend further clarification of the how the terms "environmental characteristics" and "competitive interactions" are being used in this manuscript. When the authors are talking about environmental filtering, are they only referring to abiotic conditions (e.g. TOC, sediment grain size)? Or does this also take into account biological interactions (e.g. availability/biomass of microbial food sources, pathogens such as viruses and fungi, etc.). For nematodes that co-exist, are "conserved traits" (Line 321) referring to the nematode itself (mouth type, life history) or the surrounding environment (presence of prey species or microbial food source)?

Reply: This description of what is meant by "environment" and traits is added at Line 38-42.

* The authors also do not make use of nematode-specific ecological analyses such as feeding groups (sensu Wieser 1953) or maturity index (Bongers 1990) - I am curious to know whether genera with certain life histories (e.g. epistrate feeders such as *Acantholaimus*) always show the same patterns of co-occurrence in the CCFZ, or if "rare" species are most commonly associated with a specific feeding group (e.g. 1A selective deposit feeders, that may feed on a specific bacterial strain). Although this type of analysis is not as critical as phylogenetic trees (see above), adding further results and discussion of these type of patterns would significantly strengthen the conclusions presented here.

Reply: Maturity Index, MI (Bongers, 1999):

Acantholaimus (Family: Chromadoridae): 3

Desmoscolex (Family: Desmoscolecidae): no information available for families of Order: Desmoscolecida

Halalaimus (Family: Oxystominidae): no information available for Family: Oxystominidae

Feeding types, FT (Wieser, 1953):

Acantholaimus: 2A, epigrowth feeder

Desmoscolex: 1A, selective deposit feeder

Halalaimus: 1A, selective deposit feeder

Although the suggested nematode-specific ecological indices are informative and have shown their value primarily in terrestrial/shallow-water ecosystems, we have reservations in applying these to our data for the following reasons.

With respect to MI: implementing this metric is problematic due to the paucity of family-specific information for deep-sea genera such as *Desmoscolex* and *Halalaimus*.

With respect to FT: It has recently been shown that the microbiomes of nematode species do not exhibit correlative patterns in association with different feeding types, morphology or ocean region, suggesting that generalist feeding may be more prevalent in this phylum than previously recognised (Schuelke, Pereira, Hardy, & Bik, 2018).

As such, we do not expect the addition of MI/FT analysis will reveal additional explanatory patterns in our dataset.

Minor Comments:

Line 93: change to "tubes were incubated for"

Reply: Corrected and moved to ESM.

Line 95: change to "of sterile water was added"

Reply: Corrected and moved to ESM.

Lines 118-128: DADA2 and QIIME1 are acronyms and thus should be always be capitalized

Reply: Corrected.

Line 120: You do not need to cite the specific URL of the DADA2 tutorial you used, referencing the software itself is enough. However, you should indicate which steps you used default parameters, and where you modified/customized any parameters because of specific features of your dataset.

Reply: Corrected.

Line 120: If possible, please post command Line scripts and parameters on GitHub or Figshare - this will help other people replicate your analyses in the future (e.g. for DADA2, QIIME, and RStudio analyses)

Reply: R scripts for DADA2 and all subsequent analysis have been added to ESM (Google docs).

Line 285: "I wonder if we can not remove..."  I think the authors forgot to remove this sentence from the text before submission

Reply: Indeed, this slipped through the proofreading somehow, our apologies.

Line 347: "Rarity in the CCFZ"  incomplete sentence

Reply: Indeed, this slipped through the proofreading somehow, our apologies.

Referee: 2

Comments to the Author(s)

The topic covered by this manuscript is very interesting, especially in relation to the organisms analyzed. Nonetheless, it was a bit hard to follow the analytical design, not because the analyses were complex, but rather because the sampling design is not clear enough. How many observations were analyzed in each analysis set? That was not clear to me. Some additional comments:

Reply: We have added a diagram of our experimental design in the ESM document (Fig. 1S) with clear indication of the number of replicates per Area.

1) I see no reason to use DistLM in cases where the response variable is not a distance matrix. It would be justifiable for UniFrac analysis, but not to PD/MPD ones. Furthermore, it is a bit hard to follow the analyses since sampling design is not too clear. I only got the information on the number of observations in the analyses in the Results section.

Reply: Part of the DistLM procedure is the generation of a dissimilarity matrix from the response variable, in our case Phylogenetic Diversity (PD); thus it is not necessary for the input to be in the form of a distance matrix as this is done through the DistLM itself. See below Fig. 4.2 from pg.125 (Anderson, Gorley, & Clarke, 2008).

Fig. 4.2. Schematic diagram of distance-based redundancy analysis as performed by DISTLM.

2) You analyzed the association between diversity measures and environmental gradients. That is hardly enough to uncover mechanisms driving diversity patterns. Please be less ambitious with your goals.

Reply: Our apologies, we have rephrased in a more modest way at Lines 49-53.

References:

- Aguilar-Trigueros, C. A., Rillig, M. C., & Ballhausen, M. B. (2017). Environmental Filtering Is a Relic. A Response to Cadotte and Tucker. *Trends in Ecology and Evolution*, *32*(12), 882–884. <https://doi.org/10.1016/j.tree.2017.09.013>
- Anderson, M. J., Gorley, R. N., & Clarke, K. R. (2008). Permanova+ for primer: guide to software and statistical methods: primer-E limited.
- Bongers, T. (1999). The maturity index, the evolution of nematode life history traits, adaptive radiation and cp-scaling. *Plant and Soil*, *212*(1), 13–22. <https://doi.org/10.1023/A:1004571900425>
- Cadotte, M. W., & Tucker, C. M. (2017a). Embracing the Nonindependence of the Environmental Filter: A Reply to Responses. *Trends in Ecology and Evolution*, *32*(12), 886–887. <https://doi.org/10.1016/j.tree.2017.09.015>
- Cadotte, M. W., & Tucker, C. M. (2017b). Should Environmental Filtering be Abandoned? *Trends in Ecology and Evolution*, *32*(6), 429–437. <https://doi.org/10.1016/j.tree.2017.03.004>
- Horner-Devine, M. C., Silver, J. M., Leibold, M. A., Bohannan, B. J. M., Colwell, R. K., Fuhrman, J. A., ... Smith, V. H. (2007). A comparison of taxon co-occurrence patterns for macro- and microorganisms. *Ecology*, *88*(6), 1345–1353. <https://doi.org/10.1890/06-0286>
- Schuelke, T., Pereira, T. J., Hardy, S. M., & Bik, H. M. (2018). Nematode-associated microbial taxa do not correlate with host phylogeny, geographic region or feeding morphology in marine sediment habitats. *Molecular Ecology*, *27*(8), 1930–1951. <https://doi.org/10.1111/mec.14539>
- Wieser, W. (1953). Die Beziehung zwischen Mundhohlengestalt, Ernährungsweise und Vorkommen bei freilebenden marinen Nematoden. Eine skologischen-morphologische studie. *Arkiv Fiir Zoologie*, *4*, 439–448.

Tree scale: 0.01

Figure 1: Approximately-maximum-likelihood phylogeny for genus *Acantholaimus* sequence data.

Tree scale: 0.01 \AA

Figure 2: Approximately-maximum-likelihood phylogeny for genus *Desmoscolex* sequence data.

Tree scale: 0.01

Figure 3: Approximately-maximum-likelihood phylogeny for genus *Halalaimus* sequence data.

Appendix B

Referee: 1

Comments to the Author(s).

The authors have done an excellent job at reworking the original manuscript, and the narrative is clearer and more concise in this resubmission. The discussion is now much stronger and the take-home points are clearly conveyed; I believe the results here are a very important scientific contribution to the literature focused on deep-sea mining impacts. I also appreciate the authors' efforts to incorporate more discussion of phylogenetic results and patterns (taking note of their intention to explore these aspects more fully in a future manuscript) - my original concerns and comments have been fully addressed, I had only one more minor comment in regard to a supplementary figure, as follows:

Figure S4 - The layout of this figure is confusing - could you clarify which bar charts represent the shared and unique ASVs (I could not figure this out by looking at the axis labels or caption, but it seems like there are some important patterns here).

- ➔ Modified caption: "Unique and shared Nematoda ASVs by replicate core. Main panel: number above bars indicate number of Nematoda ASVs found in Areas marked by filled circles; a column with a single filled circle and no vertical black bars signifies unique ASVs (i.e. not shared with another sampling location). Bottom left graph: total number of Nematoda ASVs in the BGR.PA, BGR.RA, IOM.C, GSR, IFREMER and APEI3 Areas."

Referee: 2

Comments to the Author(s).

My previous comments were considered in relation sampling design and goals. I still think there is no reason to use a distance-based analysis where the variable is not a distance matrix. I agree that is not wrong, but it is for sure unnecessary. That said, I have nothing more to add in this review.

- ➔ This was also a comment of referee 3. We have now replaced DistLM with multiple linear regression analysis.

Referee: 3

Macheriotou et al. investigate the prevalence of rare taxa and phylogenetic diversity of nematode assemblages in the Clarion Clipperton Fracture Zone (CCFZ). Investigation of phylogenetic diversity has only rarely been attempted in deep-sea environments, yet holds great promise for elucidating the relative importance of different factors in shaping organismal communities. Further, the CCFZ is a zone of intense scientific and industrial interest at present because of its juxtaposition of seemingly high environmental sensitivity with high commercial value. Because of this, I was excited to review this manuscript, and I believe it would have great appeal to the broad audience of Proceedings B.

The analyses of Macheriotou et al. are generally well designed and implemented. They demonstrate a prevalence of rare taxa, and suggest assemblages to be typically phylogenetically underdispersed. These results imply a high sensitivity of CCFZ nematode assemblages to environmental perturbation, and a high risk of local species extinction should assemblages be disturbed by mining activities.

I was asked to review this paper following an initial round of review, and I am generally satisfied with the authors' response to prior review comments. However, I would like to see the authors address a small number of points prior to manuscript acceptance.

In particular, I would like the authors to defend their use of DISTLM in the manuscript when the use of a generalised linear or additive model might be considered more appropriate. I would like the authors to explain why more environmental data was not available for the study (e.g. temperature, oxygen concentration, current speed, bathymetric position index, seafloor slope), and I would like a correlation matrix of the environmental data analysed to be added to the ESM.

- We have replaced the DistLM with a multiple linear regression analysis which resulted in approximately similar output and explanatory power with the exception that no significant model could be constructed for *Desmoscolex*. The general rule of thumb (based on Frank Harrell's book, Regression Modelling Strategies) is that in order to be able to detect reasonable-size effects with reasonable power, one needs 10-20 observations per parameter (covariate) estimated. Our (admittedly) limited sample size of n=23 means that including additional variables would decrease the explanatory power of the regression. We believe that we have included relevant parameters for our analysis given the amount of variation explained thereby. A correlation matrix chart has been added to ESM Fig. 5S.

I was confused by the lack of an abstract, and I suggest the authors consider adding mention of the relationship between degree of phylogenetic dispersion and sensitivity of communities to environmental change to their introduction and discussion sections.

- Abstract was included in a different field of the online submission platform; added to beginning of MS.

Title

Suggest changing to 'Clarion-Clipperton Fracture Zone' to match the rest of the manuscript.

- OK.

I cannot find an abstract in the manuscript. This this intentional? Please add an abstract.

- Added.

Introduction

L12: Change comma to semicolon.

- OK

L14: Suggest listing some reasons for increasing demand for these metals. E.g. electric car batteries.

- Modified: "Due to the increasing global demand for these metals in high-tech industries (e.g. electric car batteries, smartphones, etc) ... "

L18: Change to '(~4-5 km depth)'.

- OK

L31-32: Suggest changing to 'These deviations from randomness are thought to be the result of species interactions, habitat preferences, and/or speciation processes'.

- OK

L35-46: You might consider adding a few sentences outlining how communities structured by abiotic factors are generally more sensitive to changes in those factors than are communities structured predominantly by biotic factors.

- Added "Consequently, assemblages that exhibit environmental filtering are thus more likely to show high sensitivity to changes in the environment than those structured mainly by intra- and interspecific interactions."

L63: Write out ASV in full here.

- OK

Materials and methods

Please explain why more environmental information is not available – e.g. temperature, oxygen concentration, current speed, bathymetric position index and seafloor slope may be important factors in shaping deep-sea communities.

- Amongst the main objectives of the sampling design of the SO239 research cruise was the exploration of the suitability of metabarcoding in identifying deep-sea eukaryotes. Temperature, oxygen concentration and current speed were included in the environmental characterization of the area and were comparable between sampling sites. We chose to include those parameters which generated the largest differentiation between sites, while at the same time retaining explanatory power in our analyses; for these we did not include the aforementioned variables.

L80-82: Clearly too late for this now, but I would like to have seen environmental data and biological data originating from the same core by subsampling.

- No change.

L109: Please state that 'UpSetR' is an R package. Same for 'EcoSimR' at line 123, and 'DECIPHER' at line 135.

- OK

L141: Change to 'Phylogenetic Diversity (PD – see below)...'

- OK

L164-170: As mentioned above, I believe that a Generalised Linear/Additive Model would be a more appropriate analysis of relationships between PD and the environmental variables. I would also be interested to see a correlation matrix for the environmental variables. Cases of high multicollinearity can compromise statistical analyses for these types of models.

- DistLM was replaced with a multiple linear regression analysis for which multicollinearity between independent variables was assessed with the Pearson correlation coefficient and the Variance Inflation Factor ($VIF < 10$ indicating absence of multicollinearity). These indicated that the correlation was fair to moderate and within the permissible range for the subsequent regression analyses; correlation matrix chart added to ESM Fig. 5S.

Results

L184: I assume these are mean values and standard errors? If so, please state this explicitly.

- These are mean values and standard deviation; modified to "mean:0.377, sd:±0.095".

L186-187: Change to '...genera than IOM.C, BGR.RA and BGR.PA, while IFREMER had significantly fewer genera than BGR.RA and BGR.PA...'

- OK

L187: Why is 'Area' capitalised throughout the manuscript?

- Modified to lower case.

L199-203: Would it be possible to put directionality on these results? E.g. 'PD increased with increasing %TOC'.

- DistLM replaced with multiple linear regression analysis; results explained with directionality.

L217: '...in 3-4 out of six Areas'. Do the authors mean three or four?

- Corrected to: "in 3 and 4 out of six areas, respectively."

Discussion

I was surprised not to find any explicit comparison of your results with those of the few other studies which have investigated phylogenetic structure in the deep sea. I suggest adding a few sentences relating to this.

→ To the best of our knowledge, only 3 studies have applied this approach to deep-sea biota. Added: "From a phylogenetic perspective nematode assemblages in the CCFZ were characterised by relatedness that was higher than expected by chance (clustering), which has also been documented in deep-sea octocorals (Quattrini et al., 2017), peracarid crustaceans (Ashford et al., 2018) and bacterioplankton (Lindh et al., 2018). L280: 'In the absence of competition...'. Make this the start of a new paragraph.

→ OK

L290: 'Random co-occurrence patterns...'. Make this the start of a new paragraph.

→ OK

L297: 'An additional striking difference...'. Make this the start of a new paragraph.

→ OK

L299: '...extreme conditions'. This is a subjective term. Extreme for humans, yes, but not necessary extreme for other forms of life.

→ No change.

L306: Change to '...the presence of common genera'.

→ OK

L309: 85% by taxa, or by percent of total reads?

→ Modified to: "high degree of rarity represented by a large proportion of unique ASVs, collectively representing 85% of the entire ASV assemblage"

L338-339: 'It seems likely...'. I'm not sure I agree with your view here. Small APEIs within mining areas may be heavily impacted indirectly by mining activities. I think discussion of this is beyond the scope of the manuscript and should be removed.

→ Removed.

L380-382: This is a strange sentence with which to start your conclusion. This is certainly not the primary conclusion of your paper. Instead, I think your key conclusion is that the nematode assemblages investigated were phylogenetically clustered. In turn, this clustering suggests a high sensitivity to environmental change. You need to stress this point more.

→ Agree. Modified to "Our data show that free-living deep-sea nematodes of the CCFZ are phylogenetically clustered and predominantly structured via the influence of the environment rather than intra- and interspecific interactions; these assemblages are thus more likely to be vulnerable to environmental change, i.e. the perturbations brought about by large scale deep-sea mining operations."

References

- Ashford, O. S., Kenny, A. J., Barrio Froján, C. R. S., Bonsall, M. B., Horton, T., Brandt, A., Bird, G. J., Gerken, S., & Rogers, A. D. (2018). Phylogenetic and functional evidence suggests that deep-ocean ecosystems are highly sensitive to environmental change and direct human disturbance. *Proceedings of the Royal Society B: Biological Sciences*, 285(1884). <https://doi.org/10.1098/rspb.2018.0923>
- Lindh, M. V., Maillot, B. M., Smith, C. R., & Church, M. J. (2018). Habitat filtering of bacterioplankton communities above polymetallic nodule fields and sediments in the Clarion-Clipperton zone of the Pacific Ocean. *Environmental Microbiology Reports*, 10(2), 113–122. <https://doi.org/10.1111/1758-2229.12627>

Quattrini, A. M., Gómez, C. E., & Cordes, E. E. (2017). Environmental filtering and neutral processes shape octocoral community assembly in the deep sea. *Oecologia*, *183*(1), 221–236.
<https://doi.org/10.1007/s00442-016-3765-4>